# Field-Effect Transistor-Based Biosensors for Environmental and Agricultural Monitoring

**DOI:** 10.3390/s22114178

**Published:** 2022-05-31

**Authors:** Giulia Elli, Saleh Hamed, Mattia Petrelli, Pietro Ibba, Manuela Ciocca, Paolo Lugli, Luisa Petti

**Affiliations:** 1Faculty of Science and Technology, Free University of Bolzano-Bozen, 39100 Bolzano, Italy; saleh.hamed@natec.unibz.it (S.H.); mpetrelli@unibz.it (M.P.); pietro.ibba@unibz.it (P.I.); manuela.ciocca@unibz.it (M.C.); paolo.lugli@unibz.it (P.L.); luisa.petti@unibz.it (L.P.); 2Smart Materials, Istituto Italiano di Tecnologia, Via Morego 30, 16163 Genova, Italy; 3Competence Centre for Plant Health, Free University of Bolzano-Bozen, 39100 Bolzano, Italy

**Keywords:** bio-FETs, environmental pollutants, plant stresses, sensors, transistors, thin-film fabrication, flexible electronics

## Abstract

The precise monitoring of environmental contaminants and agricultural plant stress factors, respectively responsible for damages to our ecosystems and crop losses, has nowadays become a topic of uttermost importance. This is also highlighted by the recent introduction of the so-called “Sustainable Development Goals” of the United Nations, which aim at reducing pollutants while implementing more sustainable food production practices, leading to a reduced impact on all ecosystems. In this context, the standard methods currently used in these fields represent a sub-optimal solution, being expensive, laboratory-based techniques, and typically requiring trained personnel with high expertise. Recent advances in both biotechnology and material science have led to the emergence of new sensing (and biosensing) technologies, enabling low-cost, precise, and real-time detection. An especially interesting category of biosensors is represented by field-effect transistor-based biosensors (bio-FETs), which enable the possibility of performing in situ, continuous, selective, and sensitive measurements of a wide palette of different parameters of interest. Furthermore, bio-FETs offer the possibility of being fabricated using innovative and sustainable materials, employing various device configurations, each customized for a specific application. In the specific field of environmental and agricultural monitoring, the exploitation of these devices is particularly attractive as it paves the way to early detection and intervention strategies useful to limit, or even completely avoid negative outcomes (such as diseases to animals or ecosystems losses). This review focuses exactly on bio-FETs for environmental and agricultural monitoring, highlighting the recent and most relevant studies. First, bio-FET technology is introduced, followed by a detailed description of the the most commonly employed configurations, the available device fabrication techniques, as well as the specific materials and recognition elements. Then, examples of studies employing bio-FETs for environmental and agricultural monitoring are presented, highlighting in detail advantages and disadvantages of available examples. Finally, in the discussion, the major challenges to be overcome (e.g., short device lifetime, small sensitivity and selectivity in complex media) are critically presented. Despite the current limitations and challenges, this review clearly shows that bio-FETs are extremely promising for new and disruptive innovations in these areas and others.

## 1. Introduction

According to the United Nations (UN), 2 billion people lack safe drinking water, while 3 billion people rely on oceans for their life, and 2.37 billion people are suffering from hunger or are unable to have healthy and balanced nutrition [1]. In an attempt to reduce all these inequalities and build a better world, the UN has introduced 17 sustainable development goals (SDGs) to be addressed by 2030 [2]; these goals range from ensuring the availability of clean water and ending hunger, to the promotion of sustainable agriculture systems [2]. In particular, SDG 6 (clean water and sanitation) aims at achieving equitable access to safe and affordable drinking water, while also reducing different kinds of marine pollution [2]. In fact, one of the targets of SDG 6 (target 6.3) aims at improving water quality by “reducing pollution, eliminating dumping, and minimizing the release of hazardous chemicals” [2]. At the same time, SDG 2 (zero hunger) targets the achievement of sustainable food production practices using appropriate agricultural systems, to help increasing the productivity in agriculture, without compromising ecosystems (especially with the advent of climate change) [2]. At the same time, the achievement of sustainable food production is highly influenced by plants and crop yields.

To help achieve the aforementioned goals, it is of the utmost importance to have access to measurement methods that are able to monitor aquatic and agricultural environments in a low-cost, reliable, and continuous manner. Continuous monitoring can in fact lead to the early detection of contaminants or stresses, giving the possibility to quickly tackle the arising problem, thus increasing the chances of preserving the environment and/or improving crops yields. The presence of pollutants is in fact causing major problems to all environmental systems, ranging from waters (e.g., rivers, seas and oceans), soil (used for farming), air, to living systems, such as plants (affecting their growth and physiology) [3,4,5,6,7,8]. In fact, water bodies are nowadays polluted with a variety of contaminants, many of which are derived from human activities; pollution (together with other factors such as over-fishing and tourism) is damaging marine habitats and, as a consequence, leading to a fast decrease in the populations of marine species, as observed in the past few years [9]. At the same time, agricultural plants’ health status and growth rate are deeply dependent on their interaction with the surrounding environment, which can be either positive (e.g., nutrient uptake) or negative, inducing stress on the plants. Typically, plant stresses are divided into two main categories—defined as biotic and abiotic stress—while the former is mainly caused by plants pathogens’ (viruses, insects, bacteria), the latter is caused by environmental factors, such as drought, pollutants and soil salinity, which could decrease crop yield by up to 70 percent, thus threatening global food security [10,11].

Pollutants and their monitoring (aiming at reducing their presence in the environment) play a key role in both environmental and agricultural applications, being finally directly linked to ensuring a sustainable life on our planet. Before starting any discussion on the means of detecting pollutants, it is now important to clarify what pollutants are. Pollutants (or contaminants) are composed of different classes of materials, principally derived from human-made products discarded in the environment after use. The main pollutants are pesticides, metals and heavy metals, micro- and nanoplastics, and other types of chemicals such as surfactants, hormones, and volatile organic compounds (VOCs) [5,12,13,14,15,16]. Most of the contaminants are persistent in the environment and require a long time (many years or even centuries) to be completely degraded. Pesticides are substances used to stop, demolish, expel, or relieve harmful pests [17]. They could be grouped based on the pest type they affect; the most common ones are herbicides, which repel weeds, insecticides, which repel insects, and fungicides, which repel molds [18]. Despite their extensive use in the agricultural world, over the last few years, many pesticides have been banned in different countries, mainly due to their potential toxicity towards humans, animals, and the environment [13]. Metals, particularly heavy ones, are found in the environment mainly because of industrial processes and their discharges [19]. Heavy metals also end up in the soil, as well as in the water used for agriculture, thus posing a risk for crop growth and plants physiological activities [20].

Traditional methods to monitor environmental and agricultural plant samples are well established; however, they often involve bulky equipment typically available in centralized laboratories far from remote areas, involve a long processing time, require an expert operator, and, in some cases, are invasive and expensive. For example, the standard methods used to analyze water and plants require collecting the samples, processing them, and then conducting the analysis using physical (e.g., chromatography), chemical (e.g., ELISA, enzyme-linked immunosorbent assay), or biological (e.g., PCR, Polymerase chain reaction) approaches [21,22]. In some cases, the combination of two or more of these approaches is required, such as for the detection of pesticides, which is carried out using gas/liquid chromatography combined with mass spectroscopy [23].

One potential alternative to traditional methods is represented by sensors (and biosensors), which are small in size, could be cheap, do not require high expertise to operate, and can potentially monitor in real-time and in situ both aquatic (e.g., contaminants in water) and agricultural (e.g., biotic and abiotic stresses in plants) environments [24,25,26,27,28]. Moreover, the use of sustainable materials for their fabrication could enable the production of environmentally friendly devices, which is another extremely important aspect in reducing electronic waste, as highlighted in SDG 12 (Ensuring sustainable consumption and production patterns) [2,29]. Existing sensors for monitoring aquatic and agricultural environments can be classified into electrical and non-electrical methods [30,31]. The latter is based on optical [32], spectroscopic [23,33], and genetically encoded [34,35] methods, which typically lack selectivity and are expensive. On the other hand, electrical-based sensors mainly rely on transistor [36,37], impedance [38,39,40], resistance, capacitance [41], or electrical potential-based sensors [42,43,44], with some of these methods enabling continuous measurements with high selectivity. Some examples of these methods are microfluidic sensors or also some biological approaches, such as DNA/RNA amplification or fluorescence in situ hybridization (FISH) [31]. Important for these applications (and not only) are biosensors, which represent a specific class of sensors, able to convert a biological or chemical response (input signal) caused by the presence of the analyte, into a measurable (output) signal, thanks to the presence of a specific recognition element [45,46,47]. A biosensor is divided into two components: the recognition element and the transduction platform. The recognition element is the part of the biosensor able to specifically bind to the analyte of interest, in order to guarantee the selectivity of the biosensor’s response. Several types of recognition elements are available (e.g., enzymes, antibodies, aptamers), and the choice is mainly dictated by the specific application. The transduction platform is the part of the biosensor that converts the input signal into an electrical output signal [48]. The transduction platform can be of different kinds, some examples are piezoelectric [49] and optical [50] devices. The most interesting transduction devices for the scope of this review, however, are electrochemical ones (very briefly, there is current flowing between the electrodes of the device), which are well-studied and have several key advantages, such as a quick response time (aiming at real-time response or a response of a few seconds [51,52]) and a small size (for example, a 3 × 3 mm device [8] or a device with a 1 cm^2^ area [53]) [54].

One of the most promising classes of biosensors that uses electrochemical transduction devices are field-effect transistor-based biosensors (bio-FETs), which are vastly used and able to combine the favorable electronic characteristics of field-effect transistors (i.e., the signal amplification implied by the working principle of the transistors) with the high selectivity towards the analyte of interest, guaranteed by the presence of a suitable recognition element. Furthermore, bio-FETs can be fabricated in small sizes (for example, a 7 × 10 mm device [55]) and thus present the possibility of being integrated into portable devices. Recent advancements in micro-fabrication and printing technologies have made bio-FETs very appealing for a wide range of different fields, spanning from research to industry, clinical, diagnostics, food safety, environmental monitoring and plant sensing applications [36,37,45,56,57,58,59].

While several recent articles have reviewed the progress made in the use of biosensors for environmental [46,60,61] and agricultural monitoring [62,63,64], there is a lack of works in the literature focusing on bio-FETs and, to the best of our knowledge, there is no comprehensive review on their use in environmental and agricultural plant monitoring. The two mentioned fields of applications require interdisciplinary research, in order to develop bio-FETs to be potentially used in the environment (considering both water and plants). Moreover, as it was seen in other articles and reviews [48,63,65], it is important to work in a multidisciplinary manner to be able to allow a better comprehension of the complex and interdisciplinary problems we are facing, and to implement suitable solutions. Only with a multidisciplinary approach, fields of study that may seem very different and disconnected can be linked and become even more relevant and important when considered together. For this reason, in this review, the focus spams from environmental to agricultural monitoring, with the aim of linking the two fields (and the technological part related to them, like material science and electronic engineering) and of showing what similarities they have and how technological advancements link and influence them both.

This review highlights the state of the art and future perspectives of bio-FETs applied to both environmental and agricultural plant monitoring. In particular, the review covers bio-FET technology in Section 2, describing its working principles, the most common FET configurations, electronic materials, and fabrication and functionalization methods. The analysis of the reviewed literature on environmental and agricultural plant monitoring using bio-FETs is the focus of Section 3 and Section 4, respectively. Finally, a discussion of the state of the art and a conclusion and future perspectives are drawn in Section 5 and Section 6.

## 2. Bio-FETs

In this section, the basic working principle of a bio-FET is first highlighted (Section 2.1), with a focus on some possible configurations and on the materials employed in their fabrication (Section 2.2), as well as on the most frequently used fabrication techniques (Section 2.3). Section 2.4 is focused on the possible functionalization techniques.

### 2.1. Bio-FET Operation and Configurations

A field-effect transistor (FET) is an active device that is composed of three electrodes (i.e., source, drain, and gate), a thin insulating layer (made of a dielectric material), and a semiconducting channel, which is the active layer of the device [48,66]. The gate electrode (which in some bio-FET configurations, as explained later, can also be an external reference electrode) is insulated from the semiconducting channel and the other two electrodes by the insulating layer, while the drain and source are connected through the semiconducting material. The so-called metal-oxide-semiconductor FETs (MOSFETs), where the gate electrode is insulated from the silicon (Si) semiconducting channel by a silicon dioxide (SiO_2_) layer, are the most commonly used types of FET devices [67,68]. In MOSFETs, Si is the fundamental part as it acts not only as the semiconductor but also as the substrate (body) while also offering two of its regions to realize the source and drain electronics doping processes (i.e., the process in which impurities are added to a semiconductor to alter its electrical properties) [69]. The MOSFET semiconducting channel is in fact formed between these two regions when the right voltage is applied to the gate [65]. MOSFETs form the basis of modern electronics, as demonstrated by the outstanding technological developments in the field [67,68]. Beside the major Si mainstream, alternative semiconductor materials, which are also suitable for flexible, stretchable, and/or biocompatible substrates (i.e., the materials on which the device is “built on”), are nowadays available and used in the realization of FETs [70,71]. These alternative semiconducting and substrate materials are essential building blocks of the so-called thin-film transistors (TFTs), a special type of FETs. TFTs typically employ inorganic (e.g., amorphous metal oxides) or organic materials as semiconductors [65]. The main difference between regular MOSFETs and TFTs relies exactly on these alternative non-Si semiconductors, which are not forming the substrate (body) themselves. In fact, in TFTs, thin films of the semiconductive layer are deposited on the substrate (which can be made of a variety of materials) [65,72]. When compared to TFTs, MOSFETs have drastically higher carrier mobility (i.e., the performance metrics measuring how fast electrons or holes can move in the semiconducting material under applied electric field [73]) because of the different materials that are used. TFTs are vastly used in other applications (and not just in the biosensor field, which is the scope of this review), such as optical display systems, power transmission systems, and data transmission systems [65]. In Figure 1a, a MOSFET is depicted, where the main Si body is p-doped (lack of electrons), while the regions underneath source and drain are n-doped (surplus of electrons). In Figure 1b, a top-gate TFT is shown (i.e., the gate electrode is placed on top of the device), another possible structure has instead the gate at the bottom of the device and is defined as bottom-gate.

Despite the possibilities of having different configurations, the basic working principles of FETs are basically the same and are described in the following paragraphs. As mentioned before, there can be two types of semiconducting materials, n-type or p-type; in n-type materials, there is a surplus of negative charge carriers (i.e., electrons) in the semiconductor; on the contrary, in p-type materials, there is a lack of electrons, which can be regarded as a surplus of positive charge carriers (i.e., holes) [45,74]. When a voltage is applied to the gate electrode (in a specific direction dictated by the type of semiconductor), a flow of charge carriers (i.e., an electrical current) is allowed from the drain to the source through the semiconducting channel [45]. In more detail, the modulation of current between the source and drain is achieved through the semiconducting channel because of the field-effect mechanisms, which is the capacitive injection of carriers close to the dielectric–semiconductor interface [48,65]. In TFTs, this mechanisms is achieved by an accumulation layer and not an inversion region, like in the case of MOSFETs.

Transistor device behavior is commonly represented using its transfer and output characteristics. Transfer curves are obtained by measuring the current (drain-source, I_DS_) versus the gate-source voltage (V_GS_) while keeping a fixed drain-source voltage (V_DS_); output curves are instead obtained by measuring I_DS_ and plotting it versus V_DS_, usually for several fixed values of V_GS_. In Figure 1c,d, examples of transfer and output curves of a device with a p-type semiconductor material are shown. Changes to the gate-source voltage result in current variations over many orders of magnitude [54]; while the voltage control is essential in electronics to realize circuits, other alternative mechanisms allow different applications. In fact, besides voltage control, current variation (and amplification) can also be caused by surface effects, local electric fields, redox (reduction-oxidation) reactions, and other chemical reactions in the bulk solution [48].

As mentioned already, there are different possible configurations/architectures for FET devices, especially in the case of TFTs. Here, the most common ones that can be used for bio-FETs are highlighted and briefly compared (depicted in Figure 2). Besides the most traditional bottom-gate FETs, the others are electrolyte-gated FETs (EG-FETs) [48], electrochemical transistors (ECTs) [75], ion-sensitive FETs (ISFETs) [76], and chemically sensitive FETs (ChemFETs) [77]. As it can be seen from Figure 2, in all these configurations (except for bottom-gate), the gate electrode is separated from the rest of the structure by an electrolyte, which could be a solid polymer, an ion-gel, or, more commonly, a liquid solution [48,78,79]. An electrolyte is an electrical conducting medium that contains ions (i.e., an atom or molecule with a net electric charge) [79]. This property is important for the scope of this review because in the majority of the cases, the analyte of interest is found in a liquid solution. Additionally, this is also convenient to interface physiological solutions.

In the studies relevant for this review, when a bottom-gate FET was the employed structure, the devices were fabricated on a Si/SiO_2_ wafer (as it can be seen from Table 1); in most of these cases, the wafer (especially the Si body) acts as the gate electrode, while the dielectric material is SiO_2_. It is important to notice that in most cases a semiconducting material was deposited on the substrate to form the channel. In Figure 2a, this structure is depicted.

In EG-FETs, when a voltage is applied to the gate electrode, the ions contained in the electrolytic solution rearrange themselves, leading to the formation of two electric double layers (EDLs) at the two liquid–solid interfaces (gate electrode–electrolyte and semiconducting material–electrolyte) [48,56,80]. In this situation, the bulk electrolyte solution behaves like an insulator [81,82]. An example is presented in Figure 2b; in this case, the gate electrode is in the same plane as the source and drain, thus, this configuration is called in-plane EG-FET [48,55]. This is not always the case, as the gate electrode could also be external, such as a reference electrode, a needle, or a microwire [48,53].

In ECTs, the active material is represented mostly by a charge permeable layer, while the gate electrode (very often external to the structure) is submerged in the electrolyte solution (Figure 2c). The conductivity and the doping of the active film are changed by the injection of ions from the electrolyte into the active material by controlling V_GS_ and V_DS_ [69,75]. As opposed to EG-FETs, the doping changes in ECTs occurs on the whole volume of the channel rather than at the thin inter-facial area [75]. Thus, low voltages applied to the gate can lead to big variations in the drain current, which make this structure a strong amplifier in comparison to other FET ones.

ISFET was the first silicon-based sensor introduced by Bergveld [76]. It could be defined as a MOSFET, where the metal gate is substituted by a reference electrode and the dielectric layer is sensitive to the analyte changes, which is the main difference from the just-mentioned structures [76,83,84]. ISFETs commonly use a sensitive membrane as a dielectric (e.g., Si_3_N_4_) to detect ions in liquid solutions and are usually applied for pH sensing. In ISFETs, a surplus of charge regions are created using an electric field to increase or decrease the local conductivity. A representation of ISFETs is found in Figure 2d.

ChemFETs are considered an ISFET covered with a membrane (over the dieletric layer) that is selective to a specific ion analyte (or sometimes also gases); the permeable membrane is present at the gate interface and can be modulated by specific chemical stimulus [77,83,85]. ChemFETs are used to sense chemical concentrations in a solution [86]. A representation of a ChemFET is found in Figure 2e.

The choice of the FET structure depends mainly on: (A) the final application, such as the need to use flexible, stretchable or biocompatible materials, and (B) the analyte of interest, as some configurations are more convenient to measure pH than others, for example (e.g., ISFET).

### 2.2. Materials

Bio-FETs are made using different materials, with the choice mainly depending on the application requirements. Materials used for substrate, semiconducting channel, electrodes, dielectric, and electrolyte are introduced in this section, according to relevant studies in the context of environmental and agricultural sensing found in the literature.

In Table 1, the most relevant studies found in the literature concerning the scope of this review are shown, and the materials used for the devices are depicted (substrate, electrodes, and active material). In addition, the specific type of sensor used in the study (i.e., device configuration), the analyte of interest, and the recognition element (discussed in detail in Section 2.4) are listed.

#### 2.2.1. Substrates

For the substrate, two main categories of materials are employed: rigid and flexible materials. As it can be seen in Table 1, Si, mainly as a Si/SiO_2_ wafer [82,87,88], is the most used material for rigid substrates. For example, Le Gall and colleagues used a Si/SiO_2_ substrate to fabricate a bio-FET for the detection of herbicides [82]. Another solid substrate employed is glass, which has the advantage of being transparent. For example, Tao and colleagues used glass as the substrate in an ECT-based sensor to detect an insecticide (methyl parathion) [8]. Quartz is another example of a rigid and transparent substrate [53]. The use of a transparent substrate could be useful when analyzing complex samples/matrices and when a visual inspection of the sample is also needed. It can also be useful when applied on the surface of plant leaves, thus allowing the light to go through it and enabling the leaves to complete their photosynthesis process [89]. Glass, quartz, and Si can all withstand high operating temperatures; for example, in the case of Si, they can be higher than 1000 °C [65].

For some applications related to enhancing biocompatibility and decreasing invasiveness, the use of flexible, stretchable, and biocompatible substrates is preferred. Some examples of flexible substrates used are plastic materials such as polyimide (PI), parylene, polyethylene naphthalate (PEN), polyethylene terephthalate (PET), paper, or some more innovative organic materials [55,90,91]. Compared to Si, plastic materials can withstand lower temperatures; for example, PI has a glass transition temperature of around 360 °C [65]. One example of innovative organic materials is cotton, which was functionalized with conductive organic polymer and was inserted into plant tissues [91]. Cotton thread is composed of natural fibers, which can be easily accepted by the plants, and it is also a cheap material. Moreover, flexible substrates have the advantage of withstanding mechanical deformation, which could be useful for in vivo sensors on plants. Bischak and colleagues developed an Organic ECT (OECT) sensor based on PET substrate to monitor extracellular signals in plants [90].

#### 2.2.2. Electrodes

The electrodes are composed of conductive materials in the vast majority of cases metals. Source and drain electrodes are typically deposited on the FET substrate; however, the gate electrode can sometimes be externally introduced to the device (i.e., a needle). When it comes to the source and drain, Gold (Au) was vastly used in the reviewed literature (with a thickness in the nanometer scale, between 40 and 100 nm), and an adhesion layer of either Chromium (Cr) or Titanium (Ti) (with a smaller thickness, between 2 and 10 nm) was also always present [55,88,92,93].

As it was seen previously (Section 2.1), in some cases, a reference electrode (which is usually external to the device) is used in FET devices, acting as a gate electrode [94,95]. The use of a reference electrode is sometimes preferred because it usually has a stable and reliable performance; in fact, the reference electrode must be susceptible to small changes in pH (and not to other interfering ions or redox reactions, in the case of ISFETs) and, depending on the application, have a small size [87,96]. The most traditional and widely used reference electrode is the commercially available Ag/AgCl electrode [95,97]. However, much effort has been put into developing even less bulky reference electrodes. One solution found in the reviewed literature is the use of a dual transistor design. In the studies, they used a sensor that consisted of two identical ISFETs located on a single substrate; one ISFET was used as a working electrode, while the other served as a reference electrode [87,96,98].

#### 2.2.3. Active Materials

As said, the semiconducting channel is the active part of the device and the main classes of materials used are carbon-based, organic materials, Si, and nanowires (which are nanostructures made from different types of semiconducting materials [99]) (as shown in Table 1).

Among carbon-based materials, carbon nanotubes (CNTs) are an extremely interesting class of materials; they are cylindrical structures derived from the roll-up of a graphene sheet [100]. Semiconducting CNTs—with a typical p-type behaviour—have been used as the semiconductor material for bio-FETs for many years [48,55,101,102]. The use of CNTs as active material for the semiconducting channel is common in bio-FETs used in environmental and agricultural plants monitoring applications, as it can be seen from Table 1. The electrical properties of CNTs can be influenced by very small changes in the surrounding environment, thus enabling the realization of efficient and very sensitive biosensors (detecting the analyte even in low concentrations and/or in the presence of other interfering molecules). They also offer a large surface area, which is beneficial in the functionalization process, and they have a relatively high carrier mobility (up to 2500 cm^2^V^−1^s^−1^) [48]. For example, Park and colleagues used a CNT-based bio-FET to monitor real-time microbial activity [103]. In the bio-FET developed by Belkhamssa et al., to detect atrazine (a pesticide), CNTs were forming the semiconducting channel [88].

Graphene, which is a thin sheet of a single layer of carbon atoms (thus very strong and has good electrical properties), is also used as a semiconducting material; the main advantages of graphene are its sensitivity and scalability, and the possibility to use the devices in aqueous solutions with good and stable performances [8,104]. Graphene has quite high carrier mobility as well, more than $10^{4}$ cm^2^V^−1^s^−1^ [105]. For example, Takagiri and colleagues developed a graphene-based bio-FET to detect Cu^2+^ ions [52].

Organic semiconducting materials made of conjugated polymers are vastly used, especially in ECT-based sensors, forming the so-called OECTs. Organic polymers are compatible with low-cost fabrication techniques [106]; they are also permeable materials and suitable for bio-electronics applications [65,66]. In general, they present a low carrier mobility (0.1–10 cm^2^V^−1^s^−1^). Examples of these are Poly(3-hexylthiophene) (P3HT), poly(*N*-alkyldiketopyrrolopyrrole dithienylthieno[3,2-b]thiophene (poly(DPP-DTT)), 2,7-Dioctyl[1]benzothieno[3,2-b][1]benzothiophene (C8-BTBT), pentacene, and, the most common, poly(3,4-ethylenedioxythiophene) polystyrene sulfonate (PEDOT:PSS) [53,82,91,107,108,109]. As an example, PEDOT:PSS was the semiconducting material in an OECT developed to detect limonin [110].

Less common but still of interest is the use of silicon nanowires (SiNWs) as a semiconducting material, as done to develop sensors to detect VOCs [111,112]. SiNWs are a type of semiconducting material which is formed from a silicon precursor, either by catalyzing growth from a vapor or liquid phase or by etching of a solid [113].

Nanoparticle array materials could be used as well as an active material because of their ability to increase the sensitivity towards the analyte [114]. For example, gold nanoparticles (Au-NPs) were used in a bio-FET sensor to increase its sensitivity and the connection between the cells and the device [115].

#### 2.2.4. Dielectric Materials

A dielectric material is an insulator, which means it does not allow the flow of current. As a gate dielectric material, the most commonly used nowadays is still SiO_2_, especially because the Si/SiO_2_ wafer is still vastly used as a substrate for many FET devices (as was shown in Table 1), and it is depicted in Figure 2a. When other materials are used as the substrate, the dielectric insulating layer is mainly not present at all because of the device structure; for example, in EG-FETs, the electrolyte acts as the insulating layer when voltage is applied (as explained in Section 2.1) [48,55].

#### 2.2.5. Electrolytes

Electrolytes are very important in many bio-FET structures because they can act as the gate insulator. As mentioned before, an electrolyte could be a solid polymer, an ion-gel, or a liquid solution [48,78]. Since in many studies interesting for this review the analyte was present in a liquid solution, the used electrolyte was also a liquid. In addition, when the devices are tested in laboratory conditions, the analyte of interest is added to the electrolyte solution, and thus, the electrolyte needs to resemble the “real” final target application as much as possible. For these reasons, the most commonly used electrolyte solutions are water (such as deionized water or tap water for applications closer to real samples), PBS (phospate-buffered saline, which is an isotonic solution commonly used in biological research), or solutions containing salts (such as NaCl or KCl) dissolved in deionized water [48,55,82].

### 2.3. Fabrication Methods

In this subsection, an overview of the most common fabrication methods used to realize bio-FETs applied to environmental and agricultural monitoring is given. The focus is on the advantages and disadvantages of each presented method to point out and show when one method is preferable over the others for a specific application.

To fabricate a bio-FET, several steps are necessary. In general, films (of the semiconducting material or of electrodes) need to be deposited and structured; there are different ways to do that, and the different steps and processes are discussed in the following paragraphs.

The desired design can be patterned on the substrate using either micro-fabrication techniques (e.g., photolithography) or alternative fabrication technologies (e.g., printing).

**Table 1 sensors-22-04178-t001:** Summary of the different device configurations and materials (substrate, source/drain electrodes, gate electrodes, active material) used in the realization of bio-FETs for environmental and agricultural plant monitoring. The analyte of interest and recognition elements used by authors are also reported.

Configuration	Substrate	Source/Drain	Gate	Active Material	Recognition Element	Analyte	Ref.
Bottom-gate FET	Si/SiO_2_ wafer	Ag	Al	CNTs	Antibodies	*Salmonella*	[116]
Bottom-gate FET	Si/SiO_2_ wafer	Cr/Au	Si	CNTs	Antibodies	*Salmonella*	[92]
Bottom-gate FET	Si/SiO_2_ wafer	Ti/Au	Si	CNTs	Aptamers	*Escherichia coli*	[117]
Bottom-gate FET	Si/SiO_2_ wafer	Ti/Au	Cr/Au	CNTs	Antibodies	Domoic acid	[118]
Bottom-gate FET	Si/SiO_2_ wafer	Ti/Au	-	CNTs	Hydrogel	*Aspergillus niger* activity	[103]
Bottom-gate FET	Si/SiO_2_ wafer	Cr/Au	Si	CNTs	DNA	P-Ethylphenol	[101]
Bottom-gate FET	Si/SiO_2_ wafer	Ti/Pt	Si	CNTs	Ag-ZnOs	Methyl parathion	[119]
Bottom-gate FET	Si/SiO_2_ wafer	Ti/Au	Cr/Au	CNTs	Antibodies	Atrazine	[88]
EG-FET	PI	Cr/Au	Cr/Au planar	CNTs	Enzymes	Acetylcholine	[55]
EG-FET	Quartz	Cr/Au	Au wire	Pentacene	Antibodies	Plum Pox Virus	[53]
EG-FET	Si/SiO_2_ wafer	Ti/Au	Pt microelectrodes	Poly(DPP-DTT)	n.a.	Glyphosate and diuron	[82]
ECT	Si/SiO_2_ wafer	Ni/Au	Ag/AgCl needle	Graphene	TCA	Cu^2+^ ions	[52]
ECT	Si/SiO_2_ wafer	Au	Ag/AgCl needle	Au-NP	Cells	Cell membrane depolarization	[120]
ECT	Glass	Cr/Au	Cr/Au	Graphene	Enzymes	Trichlorfon	[51]
ECT	Glass	Cr/Au	GCE	Graphene	ZrO_2_/rGO	Methyl parathion	[8]
ECT	Si/SiO_2_ wafer	Ti/Au	Ti/Au planar	PEDOT:PSS	CNPs-SF patch	Limonin	[110]
ECT	Cotton thread	-	Ag wire	PEDOT:PSS	n.a.	Ions	[121]
ECT	PET	n.a.	Ag/AgCl needle	PBTTT + P3HT	Ion exchange gel	Extracellular signals	[90]
ECT	PEN	Ti/Au	Ti/Au planar	PEDOT:PSS	Enzymes + PtNPs	Glucose and Sucrose	[122]
ECT	PEN	Ag	Ag/AgCl planar	PEDOT:PSS	Ion-selective membrane	Potassium	[123]
ISFET	Si/SiO_2_ wafer	n.a.	n.a.	Si	Enzymes	Indole alkaloids	[87]
ISFET	Si/SiO_2_ wafer	Poly-Si/Al	Si	Si	Enzymes	Glycoalkaloids	[98]

Au-np = gold nanoparticles, PI = polyamide Ag-ZnO = Silver-zinc oxide, GCE = glassy carbon electrode, CNPs-SF = ceria nanoparticle silk fibroin, TCA = thiacalix[4]arene,
PET = Polyethylene terephthalate, PEN = Polyethylene naphthalate, ZrO_2_/rGO = zirconia/reduced graphene oxide.

Micro-fabrication is the process of fabricating miniaturized structures in the micrometer scales or even smaller. Photolithography is the process that transfers a pattern from a master (mask) to the so-called photoresist material (which is a photosensitive material dissolved in a solvent) by light exposure [65]. With this process, the desired shape of the electrodes is patterned on specific photoresistant materials through an exposure step with a so-called mask-aligner [110]. Standard photolithography techniques allow achieving a resolution in the micrometer scale. To obtain lower resolutions, some more expensive and sophisticated tools are available, such as electron beam lithography or optical projection lithography [124,125,126]. There can be two different approaches in the photolithography process; the deposition of the desired materials (such as metals for electrodes) can be performed either underneath the photoresist (so before the light exposure step) or on top of the photoresist (in this case after exposure to mask aligner). In the first case, after the photoresistant material is exposed to light (and a development step is performed to remove the photoresist where it is not needed), an etching step is performed, where the desired structure is transferred to the underlying layer. Instead, in the second case, a development and lift-off step is needed to remove the photoresist from the undesired structure so that the desired materials (such as metals for electrodes) can be later deposited on the final structure. High-quality transistors with an advanced resolution (even as low as 1–3 μm) can be obtained using this method [55,104]; however, it is expensive, and it involves complex processes. Most of the studies relevant to this review used photolithograpy (plus deposition, etching, or lift-off) as the fabrication method for the devices, especially those which employed Si/SiO_2_ as substrate material. Photolithography can be used both on rigid and flexible substrates, thus making this technique usable for different applications. Photolithography is either followed or preceded by different deposition techniques (e.g., atomic layer deposition (ALD), sputtering, evaporation, chemical vapour deposition (CVD)) of the different device layers [65]. The electrodes are very often deposited on the device through physical vapor deposition (PVD) methods, where the main ones are thermal evaporation or sputtering [65]. Very often underneath the real electrode layer there is an adhesion layer. For example, Belkhamssa and colleagues used sputtering to deposit Ti and Au to form the drain and source electrodes of their bio-FETs [88]. Diacci and colleagues, instead, used thermal evaporation to deposit metal films of Ti and Au [93]. These techniques are well developed now, especially because they have been developed since the 1960s thanks to MOSFET fabrication [67,68]; however, they also require powerful and specific equipment, and they are in general quite expensive. In any case, they are still the best solution for electrodes deposition since there is no better solution available at the moment that can outperform these methods.

On the other hand, alternative fabrication techniques (e.g., printing or laser-patterning) are cheaper, and they are suitable for large-area electronic devices; they are also good for using different kinds of substrates (including flexible ones, such as paper or plastic, but also rigid substrates, such as Si/SiO_2_ wafer) [116,127,128]. The most popular printing technique is screen-printing, which relies on transferring an ink onto a substrate through the use of a mesh; the use of a blocking stencil allows the printing of a desired pattern [123]. Despite being popular, this method suffers from bigger feature sizes with respect to photolithography [129].

The active material is often deposited through CVD methods, such as in the case of graphene [8,52] or CNTs [92,101]. In CVD methods, a chemical reaction happens on the proximity of the substrate; because of the reactants present in the vapor-phase, this leads to the formation of a solid thin film [65]. CNTs and graphene can be grown with CVD methods directly on the substrate; of course, this is possible only when the substrate can withstand the high operating temperatures [117]. In most of the other studies, the active material was grown using CVD methods but not directly on the substrate; rather, it was later transferred to the bio-FET device. Often, this transfer is carried out when graphene is the active material; polymethyl methacrylate (PMMA) is spin-coated on graphene. Next, through a wet chemical process, the graphene-PMMA layer is transferred on the substrate, then annealing at high temperature is performed with a final rinsing step to obtain the final semiconducting channel [8,51,52,130].

Spray coating (using a spray coater machine) is often used to deposit CNTs [55] or other nanomaterials, such as SiNWs [112]. Spray coating consists of an homogeneous deposition of liquid droplets onto a substrate; the solution is decomposed into small droplets at the nozzle of the spray head and then deposited onto the substrate; finally, the dispersing fluid is evaporated by heating; thus, a thin coating layer is formed [48]. For CNT deposition, some studies used a drop casting method instead, where the CNTs solution was drop casted on the device substrate at room temperature, and then the substrate was dried (either air-dried at room temperature or with the use of an oven or an N_2_ gun) [88,118,119,131].

For organic semiconducting materials, the most common deposition method that has been encountered in this literature survey was spin coating [82,90,122]. In this case, the material is applied on the center of the substrate, and then when substrate is rotated (at speed up to 10,000 rpm), the material spreads on the substrate thanks to the centrifugal force [132]. On the other hand, dip coating, which is considered one of the earliest wet chemical deposition techniques, has been used to deposit PEDOT:PSS to fabricate an OECT on a cotton thread as the substrate material [91]; such a method starts with immersion in a precursor solution, deposition of the active material, and finally ends with the evaporation of the solvent [133].

### 2.4. Functionalization Methods

Bio-FETs combine the transistor technology and a recognition element. The recognition element allows the detection of the specific target analyte, thus enhancing the selectivity of the sensor. As mentioned above, there are different mechanisms involved in current variations in FET devices (such as surface effects, local electric fields, and chemical reactions in the bulk solution); all these mechanisms are indeed very important for bio-FETs, where the interaction (binding) of the analyte to the recognition element leads to changes in the electrostatic surface potential. The interaction can thus lead to either a decrease or an increase in the measured current, which typically depends on how the recognition element rearranges its conformation upon analyte binding; these changes are used to detect the analyte of interest.

Biorecognition elements are frequently used, mostly due to their biological origin (e.g., antibodies, cells, enzymes, aptamers) and thus their biocompatibility and availability. In Table 1, the type of recognition elements used in each study is shown. Figure 3a shows a basic structure of a bio-FET. In this example, a bottom-gate FET functionalized with recognition elements is depicted, and the analyte is present in a liquid solution.

As mentioned, the most common recognition elements are antibodies, aptamers, enzymes and cells [134]. Antibodies are immunological proteins produced by the immune system and used to identify and neutralize foreign antigens (i.e., a lock and key analogy) [135]. Despite being highly accurate and specific to the analyte, antibodies suffers of being expensive and the isolation and production process is time consuming. Aptamers are single or double stranded nucleic acids (DNA or RNA), having a unique sequence which gives them a specific structure; this structure can bind to one small molecule with high selectivity [117,136]. Aptamers have a good thermostability and a quite long shelf life. As of today, also aptamers’ production is time consuming, since they are isolated through an in vitro selection process, defined as systematic evolution of ligands by exponential enrichment (SELEX) [136]. Enzymes are proteins that act as biological catalyst in biological reactions; they also present a high selectivity towards the specific target molecule and show high affinity as well [46,55,137]. During their production process, enzymes are sensitive to degradation (from pH or temperature variations), thus making the production difficult (or at least the whole process needs to be well handled) [138]. When the used enzymes are involved in reversible reactions, the binding site is not affected and thus the enzyme (and biosensors) could possibly be reused [138]; instead in case the reaction is not reversible, then the problem of non-reusability of the biosensor arises [139]. Cells are a low-cost biorecognition element thanks to their availability and simple preparation methods. However, being living systems, cells need to be in a very controlled environment, which is not always possible to achieve when developing biosensors to be used in the environment or on plants [120]. A schematic of the general structure of the just discussed recognition elements is shown in Figure 3b. There are also other less common recognition elements available; some are worth mentioning since they were used in some of the studies relevant to this review. For example, silver (Ag) nanorods decorated with Zinc oxide (ZnO) nanoparticles were used by Kumar and colleagues, since they have strong affinity with the specific pesticides they wanted to detect [119]. Hydrogel materials consist of a three-dimensional cross-linked polymer network [140]. Hydrogels have also been used, both as a non-specific recognition element [103] or as a way of functionalizing the gate device (different functionalization techniques are described later), as in the case of Le Gall et al., where *cyanobacteria* were entrapped in the hydrogel [82]. Another alternative material that was used as a recognition element was thiacalix[4]arene (TCA), which is a molecule composed of benzene rings linked via sulphide bridges [118]. Ceria nano particles (CNPs) are antioxidants and were used as recognition element to be able to monitor limonin at an extremely low concentration [110].

In bio-FETs, the device needs to be functionalized with the recognition element of choice; the functionalization can be carried out on different parts of the device, the gate electrode, the substrate surface, and most commonly the semiconducting channel [48,55,110,141,142]. Drop casting of the solution (containing the recognition elements) directly on the device is sometimes used for gate functionalization [51] and for semiconducting channel functionalization [88,103,118,131]. Takagiri and colleagues instead exploited the possibility of their recognition element (TCA) to form the π–π interaction with graphene, to functionalize the semiconducting channel [52]. Indirect immobilization can also be employed. These techniques use an additional binding molecule [48]; for example, So et al. modified the aptamers with biotin and then exploited its strong interaction with streptavidin (present in the semiconducting channel) to immobilize the aptamers on CNTs [117]. Another indirect immobilization approach was shown by Lerner and colleagues, where through wet-air oxidation, they modified CNTs to create carboxyl groups, and these groups where then exploited to bind the antibodies [92]. To functionalize the gate electrode, Le Gall et al. used an hydrogel material, where *cyanobacteria* could be entrapped and thus be in close proximity to the electrode [82].

When the analyte arrives in the proximity of the device, a specific interaction occurs between it and the recognition element; this interaction is able to influence the current that flows between drain and source electrodes. The current variation, resulting from this interaction, can be used to detect the analyte’s presence and measure its concentration [117,119].

The successful functionalization of the device with the recognition element gives the final sensor device. Afterwards, its behaviour needs to be studied in order to assess the following aspects. (a) The biosensor sensitivity (the ability to differentiate tiny differences in the concentration of the analyte); Lee et al. measured a different change in the bio-FET current when the analyte’s concentration was 52 fM compared to when the concentration was 70 fM [102]. (b) The selectivity of the sensor (the ability to differentiate the analyte of interest from other components); for example, in an ECT sensor to detect Cu^2+^ ions, in the presence of other metal ions at a concentration of 500 μM, no response was seen, while upon the addition of Cu^2+^ ions at 50 μM, a response was shown (the type of response is presented in Section 3.3) [52]. (c) The re-usability (how many times the device can be re-used); in a recent study an ISFET was developed with a lifetime of 10 to 20 measurements, the limitation was due to the decrease in the biosensor response [87]. (d) The reproducibility of the sensor response (the ability to repeat the same results, with similar sensitivity and selectivity, when a bigger number of sensors is tested); for example, Wang and colleagues fabricated three devices that presented a relative standard deviation of 1.42% [51]. (e) The limit of detection (LOD) (the minimum concentration of the analyte A that can be sensed [138], which can be calculated using the following equation, A = B + qC, where B is the mean of the blank measures, C is the standard deviation of the blank measures, and q is a numerical factor chosen according to the confidence level desired [143]). (f) The biocompatibilty of the materials used in the sensor (the capability to stay in the measured system (e.g., the plant) for a long time with minimal immune response [144]). Furthermore, finally, (g) the sensor invasiveness (the ability to minimally being harmful to the measured system when being attached to its surface [145]).

In Section 3, bio-FETs used for environmental monitoring are presented, with a focus on aquatic environments (such as rivers and seas). On the other hand, bio-FETs developed for agricultural monitoring, with a focus on plant sensors, are discussed in Section 4.

## 3. Bio-FETs in Environmental Applications

Nowadays, pollution in the environment is a cause for major concern since it does not only affect the environment itself, but also the surrounding flora and fauna. Pollution is part of the planetary crises defined by the United Nations Environment Program (UNEP), which mentions, among the major issues climate change, nature and biodiversity loss, pollution, and waste [9]. Environmental pollution itself is an important topic in the UN agenda and is also correlated with many SDGs [2]; the UN has also recently established the Pollution, Health, and Environment Unit, which has the goal of addressing the issues between environment and health [9]. Furthermore, there are many pollutants recognized as a serious global threat both to humans and ecosystems. Some examples of these pollutants are highlighted in this section.

Pesticides are persistent in many environments (water and soil especially) and can be harmful to humans; the presence of bacteria in the environment can indicate contamination (such as fecal contamination for *E. coli*); metal ions are also accumulated in the environment because of human-related activities and are considered toxic for humans; finally the presence of chemicals in general is also considered an environmental problem, since chemicals can contaminate water bodies, soil and so on. For these reasons, it is highly important to detect the presence of these contaminants in the environment. Despite several recent articles have reviewed the progress made in the use of biosensors—especially based on optical or electrochemical transduction platforms—for environmental monitoring [46,60,61], there is a lack of works focusing on bio-FETs.

In this section, we will provide an exhaustive review of bio-FETs used to detect different types of contaminants, including pesticides (Section 3.1), bacteria (Section 3.2), metal ions (Section 3.3), and other environmental contaminants (Section 3.4).

In Table 2, the most relevant studies are shown, highlighting the analyte of interest, the specific recognition element, the achieved range of detection, the application of the developed sensor, and finally, the lifetime of the devices.

### 3.1. Pesticides

Pesticides have been used since the 1950s because of their ability to repel weeds, insects, or molds, therefore leading to improved crop yields and ensured food security [17,18,148]. Pesticides have been shown to be persistent in the environment, especially in soil or water [13,17,18]. In fact, when pesticides are used, their residues often have an offsite movement towards close waterbodies or soil, thus leading to the deterioration of water and soil quality, as well as posing risks to the living organisms inhabiting those ecosystems [13,148].

Based on their chemical structure, pesticides can be divided into different classes; the most important ones are (for the scope of this review): triazine, organophosphate, urea, and glyphosate-based pesticides [149]. Triazine pesticides are primarily used as herbicides, and they influence the photosynthesis of plants. Some examples are atrazine and cyanazine [149]. Organophosphate pesticides are mainly used as insecticides. They are very toxic to bees, wildlife, and even humans since they inhibit acetylcholinesterase (AChE) in the nervous system [150]; some examples are malathion, methyl parathion, and trichlorfon [149,151]. Urea pesticides are also primarily used as herbicides, affecting the photosynthesis of plants. They are well adsorbed by the soil but less soluble in water, and they show a low acute toxicity in humans [149,152]; some examples are diuron and chlortoluron. Glyphosate-based pesticides are herbicides characterized by a broad spectrum, systematicity, and non-selectivity, which are also easily adsorbed by the soil [149].

Atrazine (ATZ) was vastly used in agriculture before being banned in Italy and many other countries [153]. It is nevertheless still persistently present in soil and water and also has drawbacks on human and animal health [154,155]. Belkhamssa and colleagues developed a bottom-gate FET for the detection of atrazine in aqueous samples [88]. The device schematic is shown in Figure 4a; they used CNTs as the semiconducting material, and the bio-FETs were functionalized with anti-ATZ antibodies. The response of the devices to different concentrations of ATZ was analyzed, based on the variation of the analytical response (ΔI ATZ); this can be seen in Figure 4a (working range of 0.001 to 10 ng/mL and an LOD of 0.001 ng/mL). The current decrease was due to the formation of an immunocomplex between ATZ and the antibodies. In spiked water samples (deionized, sea, and riverine water, where atrazine was intentionally added at two different concentrations, 0.01 and 1 ng/mL), the devices showed a good recovery percentage between 87% and 108%. However, in real water samples, the devices could not detect ATZ, which was likely due to the more complex medium that was tested and possibly to the very low concentration of ATZ in those samples. This is a main drawback for the sensor since this actually hinders the possibility to use the devices on real water samples.

Another promising device, developed by Kumar et al., consisted of a bottom-gate FET specific for the detection of methyl parathion. Here, ZnO nanoparticles decorated with silver (Ag) nanorods (Ag-ZnO) were used as the selective recognition element [119]. The devices showed good selectivity towards methyl parathion and showed an increase in measured current since the interaction between Ag-ZnO and methyl parathion affected the conductivity of the CNTs (that composed the semiconducting channel). The devices were tested in a concentration range of $1\;\times \;10^{-16}$ M to $1\;\times \;10^{-4}$ M of methyl parathion; the LOD obtained was $0.27\;\times \;10^{-6}$ M, which is lower than the allowed limit in the environment for this pesticide. They were also used in real samples; rice and soil samples were sprayed with methyl parathion, extracts were obtained from them, and the methyl parathion presence was measured with a good recovery (99% for rice and 101% for soil). The main drawback was that the real samples underwent a quite long extraction method, which involves also the use of solvents; however, the extraction method was indeed needed since the organic parts (of the soil sample) had to be removed prior to analysis, and in general, the final analyzed sample had to be in liquid form [119].

Le Gall et al. developed a hydrogel-gated EG-FET to monitor the influence of two pesticides (glyphosate and diuron) on *cyanobacteria* [82]. They entrapped photosynthetic *cyanobacteria* on the gate electrode thanks to an alginate hydrogel; the authors claim to be the first ones to use this functionalization method; thus, it is quite innovative and could be used for the functionalization of the device with more complex molecules. However, this method probably cannot be used with every type of recognition element. Photosynthetic *cyanobacteria* produce oxygen under light exposure, while they do not produce it under dark conditions [156]. They exploited this characteristic for the bio-FETs since in light periods, the oxygen produced by the *cyanobacteria* was directly electro-reduced on the gate electrode, and this resulted in a measurable gate current, which, in turn, gave rise to an amplification of the drain current. The change (decrease) in the measured drain current when glyphosate or diuron were added to the device showed that the two pesticides did indeed affect the photosynthetic activity of the *cyanobacteria* [82]. The results were promising; however, at the moment, the device lacked selectivity towards the pesticides since they only proved the fact that the pesticides had a negative effect on the *cyanobacteria*, but the effect was not pesticide-dependent.

More recently, an OECT device (with graphene as semiconducting material) was developed for the detection of trichlorfon using acetylcholinesterase as the biorecognition element (a specific enzyme) [51]. The enzyme was functionalized on the gate electrode; the biosensor was used to detect trichlorfon based on its ability to inhibit the enzyme acetylcholinesterase [157]. The device showed an increase in I_DS_ (of about 0.23 μA at 10 nM concentration) even when low concentrations of trichlorfon were present in the solution (LOD of 10 nM, range of concentration tested was 10 nM to 3 μM). The change in current was due to the inhibition of acetylcholinesterase activity caused by trichlorofon, which was causing a change in the production of thiocholine from acetylcholine chloride (which competes with trichlorofon). The device did not show any response when other interfering molecules were added (such as metal ions, glucose, or other pesticides) and could detect trichlorfon in spiked rice samples (concentrations of trichlorfon 300 nM and 3 μM) [51]. Furthermore, in this case, to test real samples, extraction and purification techniques were used, which are time-consuming and require the use of solvents. Something that seems interesting but has not been shown is the ability of the device to detect trichlorfon in real samples.

Bhatt and colleagues developed an EG-FET for the specific detection of acetylcholine (a neurotransmitter) and to detect the pesticide malathion using the sensor in an inhibition test [55]. The enzyme acetylcholinesterase was used as the biorecognition element [55]. A range of acetylcholine concentrations from 1 pM to 1 mM was tested, and an increase in current was seen with increasing concentrations (due to the rearrangement of the immunocomplex that was formed). A very good selectivity towards acetylcholine was achieved since the device was not affected by the presence of other neurotransmitter molecules (solutions with a fixed concentration of acetylcholine but different concentrations of serine and glycine, all showed the same increase in I_DS_: from −4 to −5.5 μA). More interesting for this review, in addition to these results, an inhibition test was performed, where malathion was added to the solution (two concentrations, 2 and 5 mg/mL); malathion was able to inhibit acetylcholinesterase, which could thus not bind to the acetylcholine present in the solution. The enzyme inhibition is linearly correlated with the concentration of malathion in the analyzed solution; thus, the EG-FET can be considered a biosensor for malathion detection. The inhibition test was also performed on spiked real samples (1.35 mg/mL of malathion), such as strawberry juice and tap water, and the inhibition of the enzyme was shown in this case as well [55]. This result is promising; however, the detection of malathion in even more complex environmental samples could be more challenging, but it is essential for developing a sensor that can be used in real life.

### 3.2. Bacteria and Toxins

There are many studies demonstrating bio-FETs for bacteria detection; here, we will focus on the works aiming at detecting bacteria present in the environment (thus considering bacteria as possible environmental contaminants). A large variety of environmental microorganisms are naturally present in the different ecosystems since they are essential for their stability; however, some microorganisms can enter aquatic environments (or also soil) because of human activities or industrial discharges, thus altering the natural microbial community [158]. For example, fecal pollution from urban run-off or wastewater effluents poses health risks for the population due to exposure to pathogenic bacteria or viruses [159]. In fact, fecal indicator coliform bacteria have been used to detect fecal pollution in the environment. In particular, *E. coli* is a bacterium that can be an indicator of fecal contamination in the environment [158], and for this reason, its detection is also important for environmental monitoring. *Salmonella* is another pathogenic bacterium often related to foodborne diseases. Biosensors for the detection of *Salmonella* could potentially be used in environmental monitoring, especially in monitoring water, to control and prevent possible infections. Toxins, which are microbial byproducts, are also considered environmental contaminants since their presence might be harmful to humans and animals [118,160]. For example, botulinum toxins, which are produced by *Clostridium botulinum*, are very toxic since they inhibit acetylcholine release, thus causing paralysis (in humans and animals). It seems that contaminated soils and sediments are primary environments for spores and serve as an incubation area [161,162]. Domoic acid is a marine biotoxin produced by bacteria belonging to the genus *Pseudo-nitzschia* [163] and is associated with harmful algal blooms and may contaminate seawater [118]. Harmful algal blooms occur when colonies of algae grow out of control and produce toxic or harmful effects on people, fish, shellfish, marine mammals, and birds [164].

So and colleagues developed a bottom-gate FET functionalized with *E. coli*-specific RNA aptamers for the detection and titer estimation (i.e., the concentration) of *E. coli* [117]. They were able to achieve good selectivity towards *E. coli* (no response of the device to other bacteria); however, the measured concentration ($3.1\;\times \;10^{3}$ cfu/mL) was a couple of orders of magnitude lower compared to the one measured with the traditional method ($2.7\;\times \;10^{5}$ cfu/mL) (traditional method involved an incubation step, followed by phenotyping fingerprinting) [117]. This could present a problem since the obtained result was not comparable with the other method. In fact, the interpretation of the results obtained with bio-FETs could be misleading.

A more recent study developed a disposable bottom-gate FET with thermally reduced graphene as the semiconducting material functionalized with *E. coli* antibodies, for the rapid detection of *E. coli* in water [165]. A decrease in drain current was indeed measured when *E. coli* (at different concentrations from $10^{3}$ to $10^{9}$ cfu/mL) was present in the analyzed solution. The decrease was about 0.2 μA at $10^{8}$ cfu/mL and was caused by the *E. coli*–antibodies interaction, which affected the graphene channel electrical conductivity. A threshold concentration of $10^{7}$ cfu/mL and an LOD in deionized water of $10^{3}$ cfu/mL were shown. The device showed good selectivity since the decrease in current was influenced by the presence of *E. coli* only and not *Salmonella* or *Streptococcus pneumonia*. In spiked river water, the LOD increased to $10^{4}$ cfu/mL [165]. Unfortunately, the sensor was designed for a one-time use, and to regenerate it, a buffer solution could be used, which, however, is not applicable if the sensor would be placed directly in the environment.

In Figure 4b, the device developed by Lerner and colleagues is depicted; they developed a bottom-gate FET sensor to detect *Salmonella* using specific antibodies (polyclonal anti-*Salmonella*) as biorecognition element and CNTs as semiconducting material [92]. As can be seen from the graph in the figure, when *Salmonella* was present in the analyzed solution, the measured ON current (i.e., maximum current) showed a net decrease (concentration range from $10^{3}$ to $10^{8}$ cfu/mL) because of the specific interaction of *Salmonella* and the antibodies. Additionally, a reduction in carrier mobility was correlated with higher concentrations of *Salmonella* [92]. The device showed good selectivity because this response was seen only with *Salmonella* and not with three other types of bacteria that were tested. They used a broth solution as the medium because it mirrors the complexity of a real sample, which is a good approach and may represent a good starting point for a sensor to be used in real environmental samples.

Park and colleagues developed a bottom-gate FET not to detect the presence of one specific bacterium but rather to monitor *Aspergillus niger* growth and activity [103]. The bio-FET was hybridized with malt extract agar hydrogel. The chemical properties of the hydrogel changed in the presence of microbial metabolites, and these changes could be detected through the measured current in the bio-FET. In fact, the measured current did not show any change for about 1 day (which corresponds to the lag-phase of the bacteria), and then a rapid increase was detected (from 130 to 150 nA in a couple of hours, which corresponds to rapid growth during the log phase). After this, a saturated level was maintained for 3 days (during the stationary phase of growth) [103]. The developed bio-FET can be used as an alternative method for microbial monitoring; however, it cannot be used for the detection of a specific microorganism since there was no specific recognition elements.

In Figure 4c, an example of a bottom-gate FET, completely based on CNTs, is shown. Here, the device was functionalized with either antibodies (as shown in the figure) or a peptide specifically for BoNT/E-Lc (type E light chain of botulinum neurotoxin) [102]. When the device was functionalized with the specific peptide, a decrease in conductance was measured in the presence of BoNT/E-Lc, which was due to BoNT/E-Lc cleaving (i.e., removing/cutting) seven amino acids from the peptide and thus reducing the intrinsic charges in the peptide. Additionally, a linear relationship between the change in conductance and BoNT/E-Lc concentration could be seen (analyte’s concentration range was from 60 pM to 5 nM). The interaction of the peptide was specific to BoNT/E-Lc because no change in conductance was detected when a different chain of botulinum neurotoxin (BoNT/A-Lc) was added. Similar results were obtained when the device was functionalized with antibodies, as can be seen in Figure 4c. Additionally, in this case, a linear relationship between the change in conductance and BoNT/E-Lc concentration could be seen (analyte’s concentration range was from 52 to 500 fM) [102]. The device was not tested with real samples, which would be important in order to understand whether the device would be influenced by more complex media (such as food-based or water samples).

A bottom-gate FET sensor was developed for the detection of domoic acid (DA) [118]. A linear relationship could be seen between concentrations of DA (range between 10 and 500 ng/L) and a decrease in the electrical signal (I_DS_), which was due to the formed immunocomplex (DA-antibodies) that reduced the mobility of holes and thus the measured current. The measured LOD was 10 ng/L, which was comparable to or even lower than other traditional methods. When artificial seawater was spiked with different concentrations of DA, the bio-FET could detect the right concentration and was comparable to one of the traditional methods [118]. However, real water samples could not be analyzed, meaning that improvement in the device still needs to be carried out.

### 3.3. Metals

Metals and heavy metals are naturally occurring elements that are produced from natural sources, such as volcanic eruptions and metal corrosion; for example, they are incorporated as trace elements into magmatic rocks, which come from magma cooling down [166]. However, increasing anthropogenic activities, such as mining, or industrial and agricultural activities has caused an increased accumulation of metal ions in the environment [167]. For example, waste incineration (which has drastically increased in the last century) is considered one of the main causes of Cadmium atmospheric emissions; instead, agricultural activities are the main sources of metal introduction into groundwater [166]. Metal ions have been shown to accumulate in the air, in drinking water, on plants, in animals, on soil, and on the earth’s surface; thus, they can be bioaccumulated by humans through ingestion (of animals, plants, and water) [167]. The toxicity of distinct metal ions is different, and for this reason, the tolerable limits in water (set by the World Health Organization, WHO) also vary between the different ions [167,168,169]. For example, Copper (Cu) has a limit of 1.3 mg/L and some of the effects it could have on humans are allergic reactions and kidney disorders [169]. Mercury (Hg) ions, instead, have a lower acceptable limit in water—0.001 mg/L—and can have effects on the digestive and immune system [167,169]. Since metal ions, and especially heavy metals ions, are considered threatening pollutants for the environment, the development of biosensors that can quickly and easily detect metal-contaminated samples has been greatly studied in recent years [59,170]. Many bio-FETs have been developed in this direction [167,171,172,173].

Takagiri and colleagues developed a bottom-gate FET device with graphene as the semiconducting material; TCA was functionalized on graphene as the recognition element. The functionalization (immobilization) caused TCA to rearrange in a specific structure, which was shown to specifically adsorb Cu^2+^ ions [52]. In Figure 4d, the adsorbtion of Cu^2+^ ions by TCA is shown. They showed a shift in Dirac-point voltage (V_DP_, which is the doping state of graphene [174]) with increasing concentrations of Cu^2+^ ions (shown in Figure 4d, range of concentration from 1 μM to 1 mM). The device presented high selectivity towards Cu^2+^ ions (V_DP_ shift of 120 mV) since no shift (or very low) in V_DP_ could be measured in the presence of other metal ions (such as Nickel, Cadmium, or Manganese; V_DP_ shift lower than 10 mV) [52]. This selectivity could be achieved basically thanks to the immobilization of TCA on graphene and the structural rearrangement this caused to TCA because when TCA was not immobilized, it could adsorb metal ions without selectivity. This is a starting point for potentially developing bio-FETs sensors for other metal ions as well, by exploiting different methods of functionalizing TCA.

Another approach was chosen by Wang and colleagues, who recently developed an EG-FET sensor for the detection of Cu^2+^ ions using specific aptamers as the biorecognition element and graphene as the semiconducting material [130]. The device showed a decrease in I_DS_ (415 μA without Cu^2+^ and 320 μA at Cu^2+^ concentration of 3 μM) and a gate voltage change when Cu^2+^ ions were present, with a range of detection between 10 nM and 3 μM (LOD of 10 nM). The decrease in I_DS_ and the change in gate voltage were caused by the change in the aptamers conformation because of the interaction between Cu^2+^ ions and the aptamers. The device had good selectivity towards Cu^2+^ because no change (or much smaller) was depicted in the presence of other ions, such as Ni^+^, Ca^2+^, and Co^2+^. The most interesting result was that they could measure Cu^2+^ ions in real samples—in this case, fish samples (with a measured current decrease of about 70 μA at a Cu^2+^ concentration of 3 μM).

Kim and colleagues developed a bottom-gate FET sensor to detect Hg^2+^ in aqueous samples, with CNTs as semiconducting material; they exploited the strong redox reaction happening between CNTs and Hg^2+^ [172]. In the presence of Hg^2+^ (starting from a concentration of 10 nM, going up to 1 mM), an increase in drain current was measured, with an LOD of 10 nM. This response was only seen when Hg^2+^ was added, while in the presence of other metal ions (such as Cu^2+^, Co^2+^, and others), no current increase was detected. This good selectivity, however, might have been less significant if more complex solutions would have also been tested, instead of only solutions containing metal ions.

### 3.4. Other Chemicals

In this section, some studies that developed bio-FETs to detect other types of chemicals relevant to environmental pollution, are presented and discussed.

Nowadays, there is a variety of different chemicals ranging from the previously mentioned pesticides to pharmaceutical or personal care products [175]. The release of these chemicals into the environment continues to be an alarming problem in many industrialized areas [175]. Some examples of these chemicals are hormones, surfactants, and their derivatives. Hormones, which are chemicals produced by our body as signals, are considered endocrine-disrupting chemicals (EDCs) since they interfere with the normal function of the body’s endocrine system [176]. Hormones are released into the environment mainly through industrial or domestic waste, thus ending up in water and causing perturbation in fish behaviour. Additionally, when present in soil, hormones have been shown to accumulate in the plants [177]. For example, 17β-estradiol, which is a natural estrogen, is found consistently in water surfaces and wastewater treatment plants’ effluent [178]. Surfactants are chemicals (used in detergents and cleaning products) that also end up in the environment after their use and have been shown to have toxic effects on animals and a bad impact on ecosystems as well [15]. Alkylphenols are one of the products of the primary degradation of surfactants in the environment [179].

Li and colleagues developed an EG-FET device for the detection of 17β-estradiol [104]. The device had two sensing units (a reference and a measuring unit) that shared a planar gate electrode; this configuration was chosen to remove the interferences that are present in complex environmental samples, as much as possible [104]. The measuring unit was functionalized with an aptamer specific for 17β-estradiol (and thus sensitive to the analyte and interferences), while the reference unit was functionalized with an aptamer insensitive to the analyte but instead only sensitive to the interferences. They detected 17β-estradiol, with an estimated LOD of 37.26 pM, which is appropriate for detecting 17β-estradiol in aquatic environments; pH interferences did not alter the detection and the LOD. They spiked tap water with 17β-estradiol and then tested the device’s ability to detect it in a more complex sample. In this case, the analyte was still detected by the device; however, a higher LOD was obtained (757.6 pM) [104]. These results are promising, especially regarding the possibility of testing the analyte in complex samples.

Belkhamssa et al. showed the development of a disposable bottom-gate FET sensor functionalized with anti-alkylphenol antibodies and CNTs as semiconducting materials, for the detection of NP (nonylphenol, a type of alkylphenols) [131]. They showed the detection of NP in the concentration range of 5–500 μg/L, with a linear relationship between the current change and NP concentration. The device showed good recovery values when artificial seawater spiked with different concentrations of NP was tested (concentrations from 6 to 400 μg/L and recovery rates between 90 and 119%). Moreover, the results were similar compared to the standard method used in NP detection [131]. As a downside, the authors did not test the selectivity of the device in the presence of other interfering molecules.

## 4. Bio-FETs in Agricultural Plants Applications

Modern monitoring platforms (i.e., Agriculture 4.0) [28] are composed of different core technology layers, which are the following: (a) the physical layer composed of sensors to acquire data and actuators to automate some practices using robots with specific automation mechanisms [180], sometimes also including unmanned aerial vehicles (UAVs) [26]; (b) the connectivity layer using the Internet of Things (IoT) to continuously connect the data acquired from sensors based on standard wireless communication protocols with different frequency bands, transmission ranges, data rates, and energy consumption rates [181]; (c) the cloud computing layer, which offers a platform (hardware, software, data storage, and security) service for the big data produced by the IoT [182]; (d) the data layer, which includes big data analytics, machine learning (ML) algorithms, and artificial intelligence (AI), typically playing an important role in transforming data into knowledge by aggregating, processing, and visualizing the large data matrices and maximizing the prediction output that may improve decision making [183]; (e) the decision support system layer, represented as the final layer by an application that is friendly to the end-user for decision making [28].

Sensors are considered an indispensable basic layer for the development of these platforms. In agricultural monitoring, sensors are used to monitor the soil (temperature, pH levels, pollutants, nutrients/fertilizers, moisture, conductivity, and salinity), the environmental weather (temperature, humidity, atmospheric pressure, wind speed, and wind direction), the plants (biotic and abiotic stresses, metabolites, pH, ions) but also livestock and other parameters of relevance, such as agricultural machinery. The scope of this section is to give an overview of the most interesting bio-FETs developed for plant monitoring because of the important role plants play in providing food, oxygen, shelter, chemicals, fiber, medicine, energy, and other applications related to reducing soil erosion and water draining.

Plants respond to stress in different ways, generally suffering from two types of stresses: abiotic and biotic. The plant response happens in different physiological stages. At the beginning of stress, plants start to behave differently from their normal metabolism/ behaviour, with a deviation inducing an alarming reaction represented by complex biological, chemical, and electrical signals that lead to an increase in catabolism (e.g., breaking down of metabolite compounds) over anabolism (e.g., sugar synthesis). Afterwards, plants enter in a new phase where they resist the stress by adopting repair processes. Later on, when the stress reaches its maximum, it leads to either a plant disease, low performance, or death. The final phase is represented by removing the stress, thus leading to a recovery stage [184]. All the aforementioned phases are regulated through biological, chemical, and electrical signals, which can be sensed in different plant organs using appropriate instrumentation, different techniques, or a selective biomarker, as we discussed in Section 1. It is crucial when developing a sensor for plants to consider the position in which the sensor will be applied because each organ needs a specific approach. Plants have different organs (e.g., leaf, stem, and fruit) to be monitored. Leaves are in charge of photosynthesis, organizing the transpiration rate, making up sugars, and releasing several VOCs [185]. Some fruits are important for reproducing the plants thanks to their seeds, while several kinds of fruits represent an edible part of the plant. As leaves and some fruits emit some VOCs and are in contact with the outside environment, they could be a good place to monitor gaseous VOC biomarkers or could be sensitive part to residual pesticides that can remain on their surface; thus, it is attractive to place the sensors on them. On the other hand, the stem provides the basic plant structure and is considered a network point to connect different parts of the plant using phloem and xylem tubes. These vascular tissues are filled with sap—a plant fluid that is composed of different compounds (e.g., water, nutrients and metabolites). In xylem, the sap moves in the direction from the root to the shoot, while in phloem, it moves from leaves (that are responsible for photosynthesis) to different parts of the plant. Thus, monitoring sap is very attractive thanks to the specific changes that happen in its composition while plants are under different stresses.

In Table 2, the most relevant studies of bio-FETs applied to agricultural plants are shown, focusing on the detected analyte of interest, the recognition element, the range of detection, the final application of the sensor, and the device lifetime.

### 4.1. Abiotic Stresses

Abiotic stresses are caused by non-living factors such as drought, temperature changes, nutrient deficiency (e.g., potassium), hypoxia (e.g., oxygen scarcity), light, wounds, anoxia (e.g., total oxygen absence), and salinity. These stresses sometimes are harmful and could lead to huge yield losses. Thus, monitoring it is extremely important in order to detect it early, which can lead to an early intervention [10,11]. Relevant studies to detect abiotic stresses are highlighted here.

Coppedè et al. developed an OECT made on a cotton thread substrate, called ”bioristor” (i.e., transistor used for plant physiology and biology), with the organic polymer PEDOT:PSS as the active material to monitor plant sap electrolyte content over a period of 22 days [91]. With the bio-FET, they showed a periodic change in I_DS_, which indeed corresponded to ion changes that happen during the circadian cycle of the plants. In fact, a reduction in the I_DS_ was obtained when the V_GS_ was increased, which could be attributed to the de-doping of the PEDOT:PSS by plant sap. Because of the substrate biological origin, cotton was convenient and more accepted by the plant, even if an immune response of the plant was observed around the inserted zone but without affecting the plant growth or morphology. The bioristor was used later for different applications in several studies to monitor the changes in the ionic composition of the plants’ xylem sap in real-time and then to correlate the changes to different plant stresses [107,121].

For example, Amato et al. implanted the bioristor in the trunk of olive trees (*Olea europaea*) to monitor the water flux density (WFD) and vapor pressure deficit (VPD) [107]. This device and how it was applied to the trunk are depicted in Figure 5c, with measurements performed over 10 days [107]. They showed that the response (the change in measured I_DS_ compared to I_DS0_, which was I_DS_ at V_GS_ of 0 V) was inversely proportional to the water flux density and correlated with the vapor pressure deficit. The change in this case was also attributed to the change in sap composition, which directly affects the de-doping of PEDOT:PSS.

In another study, the bioristor was integrated within the stem of tomato plants (*Solanum lycopersicum*), and it was applied to early detect drought stress [121]. The bioristor could detect changes in ion concentration within the first 30 h of water deprivation; the measurement period consisted of 23 days, and the response was measured as mentioned in the previously cited studies [121]. Furthermore, Vurro and colleagues used the bioristor to monitor xylem sap ion changes in different VPD conditions [147]. Furthermore, in this case, the studied plant was a tomato plant (*Solanum lycopersicum*). The analysis consisted of 15 days of constant measurements of the sensor response, where high and low VPD conditions were alternated. When VPD decreased from 1 to 0.7 Kpa, a rapid positive slope was seen in the response (which corresponds to the change in I_DS_). Conversely, when VPD was rapidly increased (from 0 to 0.7 Kpa), a negative slope was seen in the response [147].

Recently, the bioristor was applied to early monitoring the saline stress in *Arundo donax* for 37 days [146]. The sensor response was modulated by the plant sap ions, resulting in an increase in the V_GS_ and a decrease in I_DS_ because of the de-doping of PEDOT:PSS. All these studies that employed the bioristor are a good starting point, especially from the biocompatibility point of view of the device; however, the device did not present selectivity towards one specific analyte (ion in these cases) because of the lack of a specific recognition element.

Takemoto et al. developed a fabrication process to obtain highly transparent electrodes with a value of 90% transmittance in a region of visibility from 400 to 800 nm [89]. With these electrodes, they developed an OECT that was fabricated on a flexible parylene substrate, with C8-BTBT as the active material; the OECT was used for on-leaf monitoring of the plant’s electric potential and had a thin thickness (3 μm). The transparent electrodes enabled a biocompatible measurement of the plant’s response to dark and light (i.e., illumination condition) in *Egeria Densa* leaves without compromising the photosynthesis process [89]. Being flexible and transparent may increase its biocompatibility, which could potentially be an ideal platform to carry out other measurements on leaf.

In a 2014, Lee et al. made a highly flexible bio-FET gas sensor with an active film of CNTs and electrodes made of graphite to measure dimethyl methylphosphonate (DMMP) vapor in air [7]. Such a sensor was integrated into the surface of *D. sanderiana cv. Virens* leaf with improved adhesion and flexibility. In fact, I_DS_ was decreased when V_GS_ was applied with increasing DMMP concentrations (from 5 to 30 ppm) due to the interaction between CNTs and DMMP. The sensor is promising for non-planar substrate applications. For example, it was tested on nail, tape, and insect surfaces [7].

Bischak et al. were able to record the action potential signals of Venus flytrap (*Dionaea muscipula*) hair upon mechanical stimulation using an OECT sensor [90]. The device was based on a PET substrate, with both P3HT and PBTTT as active materials. In this study, the performance of the transistor was improved thanks to a layer of an ion exchange gel between the electrolyte and the active layer; such gel was able to uptake ions from the electrolyte and inject its ions into the active layer to dope it [90]. Upon triggering the plant hair, a change in I_DS_ was measured (from 0 to −4 mA). However, despite being promising, some ion gels could cause toxicity when applied to biological tissues; thus, the development of biocompatible gels could be a solution to this issue.

Tao et al. monitored the concentration of methyl parathion pesticide, which can be dangerous to the human nervous system (as already mentioned in Section 3.1) [8]. The sensor had an OECT configuration, with an active layer of graphene and gate electrode functionalized with zirconia/reduced graphene oxide (ZrO_2_/rGO) as a recognition element. The OECT was attached on the leaf of Napa cabbage (*Brassica rapa Pekinensis*) [8]. The pesticide measurements were conducted in vitro, and the lifetime of the sensor was 28 days. Such a sensor could be considered as a platform to monitor other pesticides; however, in vivo studies are needed to prove its efficiency in real environmental conditions.

Strand et al. developed a low-cost OECT using the screen printing technique on top of the PEN substrate, with an active layer composed of a mix of PEDOT:PSS and sorbitol to monitor potassium [123]. The output curves of the device was much enhanced when sorbitol was mixed with PEDOT:PSS, which might be due to the fact that sorbitol increases the conductivity of PEDOT:PSS [123,186]. An ion-selective membrane specific for potassium was then used to increase the device selectivity toward nutrient monitoring in plant sap; the sap was taken from different genuses of trees (e.g., Maple, Picea). The device durability was 4 months, and the tested concentration range was $10^{-3}$ to $10^{2}$ mM. In fact, measured current was increasing when potassium concentration was increased. The sensor could be used to detect potassium deficiency early, which is a stress that may disturb the plant physiology, ultimately resulting in a low yield [123]. The developed sensor could be used as a platform to monitor other ions by simply changing the selective membrane. However, despite being promising, the measurement method was invasive because of the extraction of the sap from the tree. Using non-invasive methods in the future could be a solution to such a matter.

### 4.2. Biotic Stresses

Biotic stress is caused by living organisms, such as microorganisms, insects, viruses, and other plants. These plant pathogens are one of the most common causes of crop damages, which, if detected early, could prevent serious plant disease epidemic. Relevant studies to detect biotic stresses are highlighted here.

Berto et al. developed an EG-FET biosensor to detect plum pox viruses (PPVs), plant pathogens that affect stone fruit species, causing significant losses [53,187]. The biosensor was based on quartz substrate with an organic active layer of pentacene and was functionalized with anti-PPV antibodies on the gate [53]. When the virus binded to the antibody, a change in I_DS_ was observed; in fact, the I_DS_ current decreased upon an increase in the virus concentration (with a range of 5 ng/mL to 50 μg/mL). This alteration was also visible in a change in transconductance (slope of the transfer curve). The formed immunocomplex (antibody + virus) caused a rearrangement of the antibody, which affects the flow of the current. The EG-FET was tested in vitro using *Nicotiana benthamiana* leaf extracts [53]; the response and the device configuration are shown in Figure 5b. Such a sensor had promising results that could be compared to traditional highly sensitive methods, such as the ELISA method. However, it should be applied in vivo to assess its efficiency.

Wang et al. developed a bottom-gate FET (Si substrate acted as gate electrode) and with CNTs as the active material to detect P-Ethylphenol gas, a volatile organic compound released by the fungus *Phytophthora Cactorum* [101], which infects strawberry plants (*Fragaria × ananassa*), resulting in a loss of fruit yield up to 50 percent [101]. The developed device was functionalized with a single-stranded DNA sequence (by π-π interaction between the CNTs and the nucleotide) to increase its sensitivity. In fact, they showed a small current increase (represented by a transconductance increase) when the sensor was exposed to ethyl-phenol gas compared to air exposure. Additionally, a shift in V_TH_ was seen, and these changes were attributed to the adsorption of partially charged VOC molecules by the recognition element. However, the selectivity of the device was overlapping with other gases released by the infected fruit, which is why the output data were processed using fitting and regression statistical techniques to increase the sensing precision toward specific VOCs. There were good results since they showed that the predicted concentrations were comparable with the true tested values [101].

In a similar study, Wang and colleagues developed a bottom-gate FET to detect gases released by citrus trees infected with greening disease—a disease caused by the pathogen *Candidatus Liberibacter asiaticu*, which can result in weak citrus tree growth and reduced quality of the fruit [188]. The bio-FET was functionalized with a ssDNA sequence specific to phenylacetaldehyde (which is a VOC biomarker for the greening disease). In addition, the maximum measured I_DS_ decreased when the device was exposed to phenylacetaldehyde compared to when it was exposed to air (4 μA compared to 4.5 μA) [188]. As a downside, the device showed a change in response for other gases (ethylhexanol, linalool, and tetradecene) as well. This low selectivity could, however, hinder the outcome of a possible real-time or in vivo study employing this bio-FET.

Another way to detect the same greening disease was developed by Saraf et al., who aimed to detect limonin, an indirect indicator of the infection [110]. The OECT sensor tested Hamlin orange (*Citrus X sinensis*) fruit extracts in order to measure abnormal limonin levels. The device was deposited with a PEDOT:PSS conductive layer by drop casting and later functionalized with CNPs as the recognition element. CNPs undergo oxidation in the presence of limonin; thus, this redox causes a de-doping of the PEDOT:PSS channel, which then results in an increase in I_DS_ [110]. The bio-FET was tested at different limonin concentrations ($10^{-8}$–$10^{-6}$ M), and indeed, an increase in I_DS_ could be seen (with an LOD of 10 nM); at 10 nM, I_DS_ was 75 μA, while at 10 μM, I_DS_ was 110 μA. The aforementioned three studies have not yet been applied in vivo. If they prove to be successful, this could be a breakthrough in agricultural research that may lead to the early detection of fruit and plant diseases using highly selective sensors.

Finally, an OECT biosensor was developed to detect plant viruses in real-time, relying on the cell membrane depolarization that happens in a single Chlorella cell (which belongs to plantae kingdom) upon the infection by a *Paramecium bursaria* chlorella virus 1 (PBCV-1) [120]. Such a sensor was based on a Si substrate and activated with gold nanoparticles necklace array, which linked the functionalized sensor with algae cells in a non-invasive way. In fact, I_DS_ increased once the virus infected the algae cell. This study can open new horizons in precisely studying virus mechanisms upon entering and exiting its host cell, allowing virologists to understand virus behaviour [120]. The device configuration and the current changes upon the virus infection are depicted in Figure 5d.

### 4.3. Plant Metabolites and pH Measurements

Plants metabolite levels change when they are under different statuses (e.g., stress, ripening) as a physiological response in order to overcome or adapt to new conditions. Monitoring these kind of metabolites may be an indicator for detecting plant health early, even before visual symptoms appear [189].

In one of the most related studies, Diacci et al. designed an OECT made of a PEN substrate with an active layer of PEDOT:PSS [93]. The sensor was functionalized with the enzyme glucose oxidase in order to be selective towards glucose, which is a signaling molecule. The sensor response was assessed in real-time in chloroplasts (plant organelles responsible for photosynthesis) [93]. Chloroplasts were isolated and extracted from tobacco (*Nicotiana tabacum*) leaves. Moreover, two different chloroplast solutions were used; one was extracted during the day and the other one during the night. The device showed an operation range of $700\;\times \;10^{-3}$–5 mM glucose concentration. The authors claimed to observe a glucose export (thus a higher glucose concentration) in a dark environment, represented by a change in the current I_DS_, compared to the chloroplast response under light conditions (I_DS_ −0.3 mA vs. −0.35 mA). Such observation is in line with plant physiology behaviour; it would be interesting to monitor other metabolites in real-time [93,190].

The same authors have developed an OECT to be applied on the stem of the Aspen tree (*Populus tremuloides*), as shown in Figure 5a [122]. The goal was to use the sensor for continuously monitoring the plants in vivo. For the first time, both glucose and sucrose were monitored in the xylem sap, with detection in the concentration range of $10^{-2}$–1 mM. In fact, the sucrose concentration behaviour, represented by the sensor’s response, was higher during the day in comparison to night. The authors used an OECT configuration made on a PEN substrate and functionalized with a mix of three enzymes (invertase, mutarotase, and glucose oxidase) on the gate electrode [122]. Such a study could open a new field of research to monitor other plant metabolites in vivo in order to understand plant circadian patterns and plant physiology in different stresses. However, the plant’s immune response upon the insertion of the sensor, which is represented mainly by the formation of cork tissue around it, can hinder the lifetime of the measurements; thus, it could be important to develop biocompatible electronic materials with minimal invasiveness.

Arkhypova et al. developed an ISFET, functionalized with acetylcholinesterase enzyme, to measure the content of indole alkaloids, a metabolite of *Rauwolfia serpentina* tissue culture, which is used in the pharmaceutical and biotechnological industry [87]. Indole alkaloids are inhibitors of acetylcholinesterase, so this study exploited this ability (similarly to what was carried out by Wang and colleagues and presented in Section 3.1 [51]). When both indole alkaloids and acetylcholine are present, the measured current is lower compared to when only acetylcholine is present because indole alakaloids also bind to acetylcholinesterase and block the reaction. They showed an inhibition level (in %), which was linear with the increasing concentration of indole alkaloids, with a concentration range of 2 to 15 μg/mL and an LOD of 0.5 μg/mL. Such a sensor claimed to be a cheap alternative to existing traditional analytical methods, which could be applied to notify the producer of such metabolites to control their growth process [87]. A good thing about this ISFET was that, with a simple washing procedure, the device could become re-usable again.

The same author had previously made an ISFET sensor functionalized with butyryl cholinesterase enzymes in order to detect glycoalkaloids, a poisonous substance found in potatoes (*Solanum tuberosum*) [139,191]. They showed a linear response in the concentration range of 2 ×$10^{-4}$–$10^{-1}$ M [139,191]. In a later study, the sensitivity of the ISFET sensor towards the glycoalkaloids was improved by using phosphotriesterase enzymes, with a resulting linear range of $10^{-6}$–$10^{-5}$ M [98].

One of the earliest studies carried out by Herrmann et al. used an ISFET to measure the pH of various tree species in vivo (e.g., *Populus balsamifera* L., *Aesculus hippocastanum* L. branches xylem). The authors developed an ISFET with an ion-sensitive membrane layer on top of the Si substrate [192]. However, being an invasive method, the authors were not sure whether the sensor’s response was affected by a wounding response in the trees. Such uncertainty could be the reason behind the limits in plant-sensing research at that time, and the advent of new technologies (e.g., thin film transistors) could be a possible reason to adopt such measurements in the future [192].

Izumi et al. fabricated an ISFET sensor to monitor the phloem sap’s pH in cucumber (*Cucumis sativus*) stem [193]. The range of pH levels monitored was 4.01–9.18. I_DS_ was measured using different pH control solutions; it could be seen that at a lower pH (4.01), the I_DS_ was higher. To calculate the sensitivity of the sensor, the V_GS_ change per 1 pH was derived from the I_DS_–V_GS_ characteristics, and the sensitivity was obtained as 40.2 mV/pH [193]. The device was later combined with electrical conductivity and thermal sensors to monitor the xylem sap in order to correlate its response signals with the pH and thus monitor some parameters of plant health in relation to soil fertilizer content [193].

Finally, a ChemFET was functionalized with a nitrate-specific ion selective membrane as the recogniton element and then was coated with poly(2-hydroxyethyl methacrylate (poly-HEMA), a material used against scratching and weathering, in order to measure nitrate in the stem of corn (*Zea mays*) [86]. In fact, I_DS_ decreased when the device was exposed to higher nitrate concentrations [86]. The measurements were carried out for over 150 h, and a linear detection in the concentration range of 0.1 to 1000 ppm was seen. The authors claimed that the sensor was minimally invasive, thanks to its ability to sense while being partially inside the plant stalk. However, it has not been proven biologically if a plant’s immune response was induced upon the insertion of the sensor, so this requires further investigations [86].

## 5. Discussion

As shown in this review, one of the main problems in monitoring environmental and agricultural plants is the complexity of the samples in which the target analyte is found. Different authors have tried many approaches in an attempt to overcome this problem and fabricate selective and sensitive bio-FETs. Considering the materials used, for environmental monitoring, most of the studies employed rigid substrates [51,52,82,88,92,102,104,111,117,118,119,130,131,165,172]. This was probably due to the fact that these materials are more known; thus, the fabrication process is more consolidated. Moreover, in the majority of these cases, there was no particular need for flexible substrates since none of the reviewed literature actually used the devices in the environment itself; rather, the devices were always used in laboratory conditions. For agricultural plant monitoring, the type of materials selected was highly influenced by the type of study and its application. For this reason, rigid substrates were mainly used for in vitro studies [8,53,110,120,139]; however, some early studies applied rigid substrates in vivo [97,192] to help with the insertion of the sensor into the plant stem [86,192,193]. On the other hand, flexible substrates were mainly used for in vivo sensing, maybe because these flexible materials are more acceptable for the plants, especially materials of biological origins [89,91,107,121,122,146,147]. Additionally, in this case, very few studies applied sensors based on flexible substrates for in vitro measurements [93,123]. Flexible substrates were used both on the stem [91,107,121,122,146,147] and on the leaf [89,90]. In fact, in terms of plant compatibility, the sensor’s adhesion to the plant’s leaf surface was improved when flexible materials were used [89]. The immune response was likely correlated with the used substrate. For example, flexible substrates stimulated lower immune responses when inserted into the plant stem [91,107,121,147] compared to when more rigid substrates were used [122]. In fact, flexible sensors have been adopted widely in other fields, particular in healthcare research (e.g., wearable sensors) [194].

In terms of the type of transistor and its relation to the analyte, ISFET was mainly used for pH sensing since this technology is well-established [97,192,193,195]. However, Arkhypova et al. applied ISFET for toxin detection [191]. ECTs (especially OECTs) were mainly used for agricultural plants monitoring, in particular to analyze ions in plant sap [91] or metabolites by functionalizing the gate with specific enzymes [122]. Only a few studies employed ECT sensors for the detection of environmental contaminants because the need to use organic semiconductors (so using an OECT sensor) was less important compared to agricultural plant applications, in which an organic active material could be more convenient to apply with biological systems. In fact, ECTs were used to detect Cu^2+^ ions by Takagiri and colleagues [52] or to monitor trichlorofon concentration [51]. For environmental monitoring, the more classical configuration of bio-FET (the bottom-gate FET) was mostly used in the found literature [52,88,92,103,111,117,118,119,131]; this was likely carried out because the devices were tested in laboratory conditions, so there were no specific needs for device configuration, and thus, the more developed methods were chosen. Fewer studies chose the EG-FET as the device configuration [55,82]. Le Gall et al. used an interesting approach where the gate electrode was functionalized with a hydrogel material in order to entrap *cyanobacteria*, whose activity was of interest for the study [82]. This entrapment method was quite innovative, and it did not affect *cyanobacteria* at all, which is important and not always easy to achieve when working with cells.

In terms of active materials, some were used for specific purposes; for example, C8-BTBT was selected in order to form an organic bio-FET because of its high optical bandgap, allowing the sensor to operate better under different visible lighting [89,196,197]. CNTs and other organic polymers proved to be convenient active materials for plant biology. For example, they were embedded in plant leaves, xylem, and roots without a negative effect [91,198,199,200,201]. CNTs were also very often used in environmental monitoring studies [55,88,92,102,103,116,117,118,119,131] since they have unique electrical properties and, according to some studies, are biocompatible [202] (which is very important when the biosensor needs to be used in field applications). However, it has been suggested that CNTs are carcinogenic when inhaled (e.g., during fabrication processes) [203]. Furthermore, their elevated surface area is helpful when it comes to device functionalization and sensitivity improvement [48,101]. It was also seen that graphene (as an active material) applied to bio-FET is a very interesting material because its properties allow the devices to detect molecules with high sensitivity. Moreover, because of its capacity to absorb different biological molecules by electrostatic interactions and elevated mobility π-π stacking, different types of recognition elements can be functionalized on graphene, thus allowing the biosensors to aim at good selectivity as well [52,104,204].

The main drawbacks highlighted for many studies include that the devices were only tested in laboratory conditions; thus, we are missing some crucial information on how the devices would behave in real conditions (analyzing real environmental samples or plants in vivo). In fact, in some cases, environmental samples were not tested at all [51,55,92,102,104,111,117,131,165,172], while in other cases, the devices were tested in real samples as well, such as river or tap water, but no results could be seen [88,118]. Wang et al. could use their developed bio-FET in detecting Cu^2+^ ions in fish samples. Wang et al.’s study was the only study reviewed that was able to achieve this result [130]. However, many studies did show some good results when spiked complex media were tested (such as spiked tap water, strawberry juice, and river water), which is a good starting point to test the real selectivity and performance of the device [51,55,88,104,119].

As discussed, the presence of interference is what limits the results of many of the developed sensors. To overcome this problem, a promising approach was chosen by Li et al. They developed a differential design, where two separate units were present in the bio-FET device, and one of them was the reference unit, which had the main objective to mitigate the effects of non-specific interference and thus improve the selectivity of the bio-FET for the target analyte [104].

Another way of enhancing the selectivity of the biosensors is the coupling of biosensors with AI and/or ML algorithms to make predictions based on large sets of data [205]. AI biosensors have three main components, which can be defined as information collection, signal conversion, and AI data processing [205]. Some examples of AI data post-processing are data classification or data modeling algorithms [205]. ML, instead, consists of computational algorithms designed to emulate human intelligence by learning from the environment [206,207]. In the literature, it was seen that some studies have started implementing AI or ML in biosensors to detect chemicals, especially gases [112,208,209]. ML can also be useful in optimizing several variables affecting the device performance and choosing the optimal conditions to fabricate a high-performance device. As an example, Paska and colleagues developed a bio-FET device (with SiNWs as semiconducting material) for the detection of nonpolar VOCs [111]. The SiNW surface was modified, forming a silane monolayer that acted as the recognition element, where nonpolar VOCs could be adsorbed. This preliminary study showed good results, even though they did not test real samples from the environment. To improve some limitations of the just-mentioned study, Wang and colleagues coupled AI with the just-mentioned bio-FET device to detect VOCs in the gas phase [112]. Some of the device parameters (such as ON current and threshold voltage) were controlled through artificial neuron network (ANN) models in order to improve the selectivity towards VOCs. In the future, this kind of approach could become useful in the field of agricultural and environmental monitoring and more.

In the future, bio-FETs could be further developed to be used on plants to indirectly monitor the environment that surrounds them. The concept, despite being applied in most cases using non-transistor-based sensors techniques, was proven, ranging from the detection of heavy metals to pesticides [210,211]. This concept could be feasible in the future in the case of bio-FETs, especially with the advent of plant wearables [145].

In conclusion, bio-FETs can enhance the selectivity of the measurements by detecting particular analytes thanks to the specific recognition elements (both of biological and non-biological origin). In addition, the advent of thin film transistors allows the use of biodegradable, biocompatible, and, in general, sustainable materials, which could be less invasive to the environment and more accepted by the plants. However, regardless of all the aforementioned advantages, sensor performance could be influenced by various external parameters, especially when the analyte is found in complex matrices and interference is present. One of the major problems encountered was the limited lifetime of the functionalized sensor, leading to decreased re-usability of the device. For example, an irreversible inhibition of the immobilized enzymes may lead to the need to re-activate or reload the enzymatic layer [138,139]. Other drawbacks have been reported on how biorecognition elements, such as aptamers and nucleic acids, could increase the sensor fabrication cost, despite improving the device selectivity [138].

## 6. Conclusions

In this review, bio-FETs and their use in environmental and agricultural monitoring are presented. First, bio-FETs are discussed from a technological point of view; in fact, the working principles, the materials used, and the fabrication techniques are all shown to understand what is important in the development of a new bio-FET device for these fields of application. Furthermore, the most relevant bio-FETs developed for the detection of contaminants or plant stresses are discussed. A variety of different bio-FETs have been reported, and their applications ranged from pesticides to bacteria detection, as well as monitoring the ion concentration in plants.

What is clear from the studied literature is that at the moment, there are still some challenges to applying bio-FETs to the environment or plants. Bio-FETs have the potential to have high selectivity towards the analyte of interest; however, this has proven often hard to achieve because of the complexity of the studied samples (e.g., plants’ tissues or seawater). Another important aspect that is often hard to achieve is the lifetime of the device, which ideally should be very long (months or even years) when the final application is to monitor in situ. This is still one of the main challenges that researchers are facing nowadays.

However, the potential of such sensors is very promising and could lead to breakthroughs in the environment and agriculture monitoring field. Overcoming the mentioned challenges will then present the possibility of using bio-FETs for in situ and fast measurements, thus removing the need for centralized laboratories and big pieces of equipment. Finally, their ability to be fabricated on flexible, biocompatible, and even biodegradable materials is also important, thanks to the advances in material sciences, fabrication, and cheap printing techniques.

## Figures and Tables

**Figure 1 sensors-22-04178-f001:**
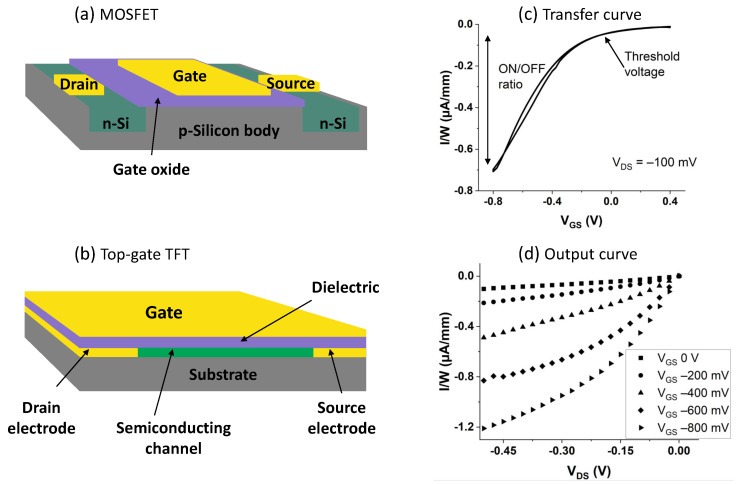
(**a**) MOSFET with the main p-Si body and two n-Si regions. (**b**) Top-gate TFT on a substrate that could be many type of materials (both rigid or flexible) and the other layers deposited on top of it. (**c**) Transfer curve of a FET presenting a p-type behavior; I/W is the measured current divided by the channel width. ON/OFF ratio is I_ON_/I_OFF_; threshold voltage (V_TH_) is the V_GS_ at which FET turns on. (**d**) Output curves of a p-type FET-device; curves were obtained at different fixed V_GS_.

**Figure 2 sensors-22-04178-f002:**
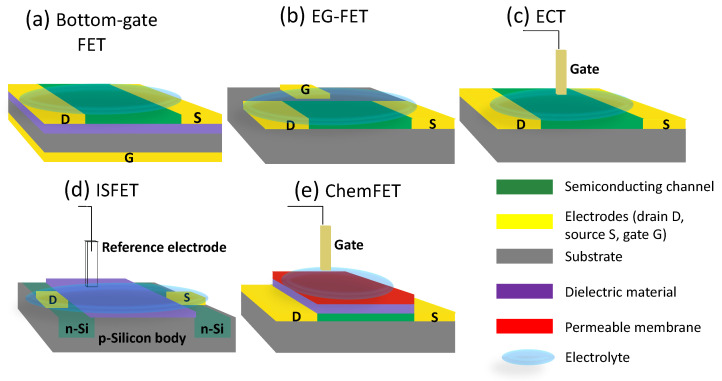
Some of the most common structures of FET devices that are used as biosensors, i.e., as bio-FETs. All structures have source and drain electrodes (in yellow), a semiconducting channel (in green), and a substrate (in gray). (**a**) Bottom-gate FET; the substrate is Si, while the dielectric material is SiO_2_. (**b**) EG-FET (planar configuration); gate is in-plane with source/drain and is insulated through the electrolyte solution. (**c**) ECT; the gate is an external electrode and is dipped in the electrolyte solution. (**d**) ISFET; the gate electrode is replaced by a reference electrode (very often an Ag/AgCl electrode). (**e**) ChemFET; the gate electrode is separated from the source and drain by an electrolytic solution, and a semi-permeable membrane is present at the gate interface.

**Figure 3 sensors-22-04178-f003:**
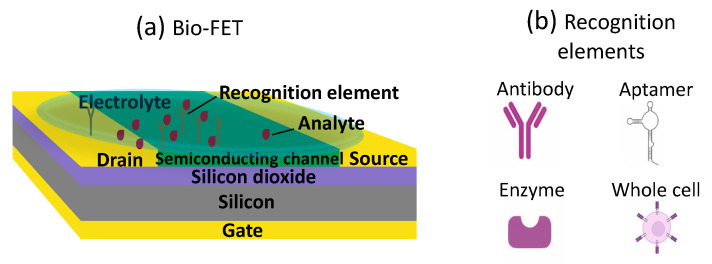
(**a**) General configuration of a bio-FET: the semiconducting channel is functionalized with a recognition element that binds to its specific analyte. In this example, the analyte of interest is present in a liquid solution; the recognition element is drawn in a Y shape typical of antibodies, which are just one example of recognition elements. (**b**) Four examples of the most used recognition elements, which are antibodies (immunological proteins), aptamers (single or double stranded DNA or RNA), enzymes (biological proteins), and whole cells. Image re-adapted from [134] (part (**b**)).

**Figure 4 sensors-22-04178-f004:**
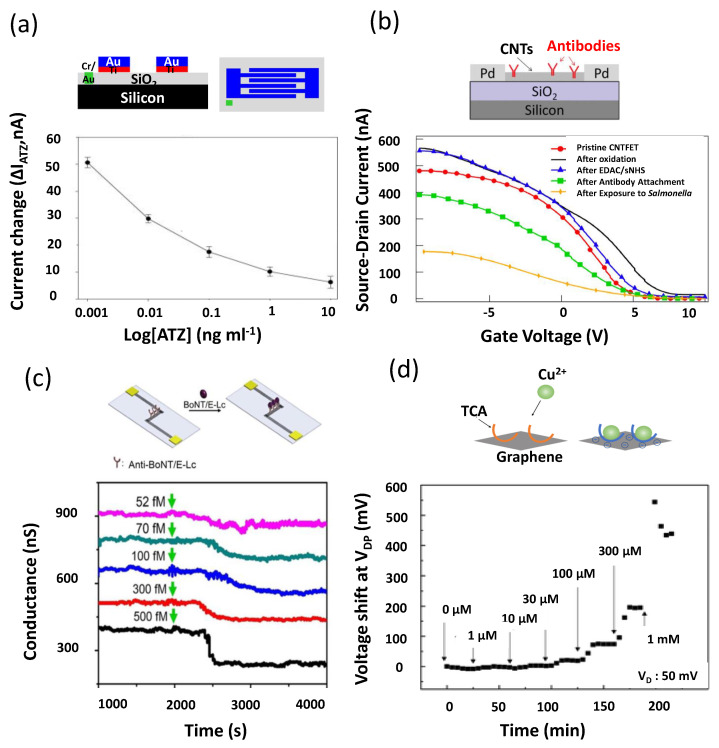
Examples of bio-FETs developed for environmental monitoring. (**a**) Bottom-gate FET to detect atrazine (ATZ). Top: cross and top view of device (CNTs were functionalized on top, not shown in the figure). Bottom: analytical response obtained at different concentrations of ATZ. The change in current decreased with increasing concentrations of ATZ because of the immunocomplex formed between ATZ and antibodies; the device showed a change in current also in the nanomolar range, and the LOD was lower than the legal limits for ATZ in food and drinking water (0.1 ng/mL). Re-adapted with permission from [88]. Copyright 2022, Elsevier. (**b**) Bottom-gate FET to detect *Salmonella*. Top: cross view of final bio-FET. Bottom: Transfer curves taken after each step of the functionalization. The net decrease in I_DS_ for the last curve (yellow line) was due to the exposure to $10^{7}$ cfu/mL of *Salmonella*, which specifically interacted with the antibodies present on the device. Re-adapted with permission from [92]. (**c**) Bottom-gate FET to detect botulinum neurotoxin (BoNT/E-Lc). Top: illustration of BoNT/E-Lc binding with Anti-BoNT/E-Lc. Bottom: real-time conductance measurement with varying concentrations of BoNT/E-Lc (ranging fom 52 to 500 fM). With increasing concentration of the analyte, the measured conductance decreased, and it reached a saturation after around 40 min. Re-adapted with permission from [102]. (**d**) OECT to detect Cu^2+^ ions. Top: schematic images of Cu^2+^ ions before and after coordinating with TCA (recognition element). Bottom: Shifts in V_DP_ (Dirac-point Voltage) with time for various Cu^2+^ ion concentrations. A small shift could be seen starting from 30 μM, while the shift started to be evident at a concentration of 100 μM and higher. Re-adapted with permission from [52].

**Figure 5 sensors-22-04178-f005:**
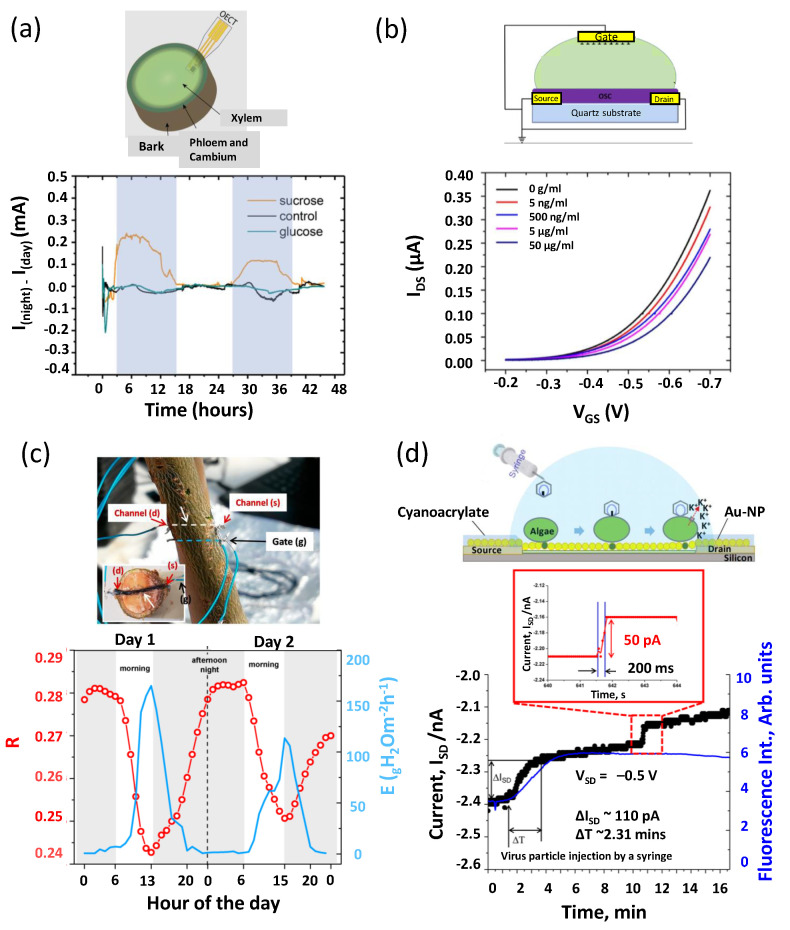
Examples of bio-FETs developed for plant monitoring. (**a**) OECT developed to monitor plant xylem metabolites. Top: device setup; the sensor was inserted into the xylem of the tree stem. Bottom: Instantaneous monitoring of sucrose (orange), glucose (cyan), and control (black) for 2 days. Bright areas correspond to daytime, while dark areas are related to nighttime. Re-adapted with permission from [122]. Copyright 2022, Elsevier. (**b**) EG-FET for the detection of plant viruses. Top: Cross-view of final device. Bottom: Transfer characteristics of the sensor after being exposed to various plum pox virus (PPV) concentrations (from 0 to 50 μg/mL); a decrease in current is shown with increasing concentrations of PPV. Re-adapted with permission from [53]. Copyright 2022, Elsevier. (**c**) OECT for the detection of ions in olive trees. Top: a representation of the sensor setup inserted into the stem of an olive tree. Bottom: diurnal fluctuations of the sensor response R (o) and plant transpiration (E) (cyan line) sensed for 48 h. Re-adapted with permission from [107]. (**d**) Bottom-gate FET for monitoring microalgae membrane depolarization. Top: representation of the sensor functionalized with algae cells and infected with viruses introduced with syringe. Bottom: the current changes upon virus infection. Reprinted with permission from [120]. Copyright 2022 American Chemical Society.

**Table 2 sensors-22-04178-t002:** Examples of recent studies that employed bio-FETs for detection of environmental contaminants and for agricultural plants monitoring applications.

Analyte	Recognition Element	Range of Detection	Application	Device Lifetime	Ref.
Atrazine	Anti-atrazine antibodies	$1\;\times \;10^{-3}$–10 μg/mL	Detection in aqueous samples	Disposable	[88]
Acetylcholine	Acetylcholinesterase	$1\;\times \;10^{-9}$–1 mM	Malathion inhibition sensing	n.a.	[55]
Glyphosate—diuron	Cyanobacteria	$1\;\times \;10^{-5}$ mM	Pesticides influence on cyanobacteria activity	Few hours	[82]
Methyl parathion	Ag-ZnOs	$1\;\times \;10^{-13}$–0.1 mM	Detection in rice and soil	35 days	[119]
*Salmonella*	Anti-*Salmonella* antibodies	$10^{3}-10^{8}$ cfu/mL	Detection in complex nutrient broth	Disposable	[92]
*Aspergillus niger*	Malt extract agar hydrogel	n.a.	Real-time monitoring of microbial growth/activity	3 days	[103]
*Escherichia coli*	RNA-based *E. coli* aptamers	n.a.	Detection and titer estimation	n.a.	[117]
*Salmonella* infantis	Anti-*Salmonella* antibodies	100–500 cfu/mL	Fast detection in solution	24 h	[116]
$Cu^{2+}$ ions	TCA	$1\;\times \;10^{-3}$–1 mM	Selective detection	Few hours	[52]
Domoic acid	Anti-DA antibodies	10–$5\;\times \;10^{3}$ μg/mL	Detection in spiked artificial seawater	Disposable	[118]
BoNT	Anti-BoNT/E-Lc antibodies-peptides	$5\;\times \;10^{-11}$–$5\;\times \;10^{-10}$ mM	Real-time monitoring of toxin	n.a.	[102]
Ions	n.a.	n.a.	Detection of WFD, VPD and light	10 days	[107]
Indole alkaloids	Acetylcholinesterase	2–15 (μg/mL)	Indole alkaloids detection	10 to 20 measurements	[87]
Glucose and Sucrose	Invertase, mutarotase and glucose oxidase	$10^{-2}$–1 mM	Metabolite monitoring	2 days	[122]
Ions	n.a.	n.a.	Measuring saline stress	37 days	[146]
Potassium	Potassium-specific ion selective membrane	$10^{-3}$–$10^{2}$ mM	Nutrients detection	4 months	[123]
Methyl parathion	ZrO_2_/rGO	$10^{-5}$–10 (μg/mL)	Pesticide detection	28 days	[8]
Action potential	Ion exchange gel	n.a.	Recording extracellular signals	n.a.	[90]
Glucose	Glucose oxidase	$700\;\times \;10^{-3}$–5 mM	Signaling molecule monitoring	n.a.	[93]
Leaf electric potential	n.a.	n.a.	Plant response to dark and light	n.a.	[89]
p-Ethylphenol	ssDNA	n.a.	Plant pathogen identification	n.a.	[101]
Ions	n.a.	n.a.	Drought stress	23 days	[121]
Nitrate	Nitrate-specific ion selective membrane	0.1–1000 ppm	Nutrient concentration detection	160 h	[86]
Ions	n.a.	n.a.	Vapor Pressure Deficit	15 days	[147]

Ag-ZnOs = silver-zinc oxide, TCA = Thiacalix[4]-arene, BoNT = botulinum neurotoxins, DA = domoic acid, WFD = Water Flux Density, VPD = Vapor Pressure Deficit, ZrO_2_/rGO =
zirconia/reduced graphene oxide, ssDNA = single-stranded Deoxyribonucleic acid.

## Data Availability

No new data were created or analyzed in this study. Data sharing is not applicable to this article.

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
