# Peer review of "Field-Effect Transistor-Based Biosensors for Environmental and Agricultural Monitoring"

_sensors, 2022, doi:10.3390/s22114178_

Round 1
Reviewer 1 Report
Please see the attachment

Author Response
Dear reviewer,
We would like to thank you for your revision and useful comments. Please see the attachment for a point-to-point reply.

Reviewer 2 Report
See attached file

Author Response
Dear reviewer,
We kindly thank you for the useful comments on our manuscript. Please see attachment for a point-to-point reply to your comments.

Reviewer 3 Report
This paper presents studies employing bio-FETs for environmental and agricultural plants monitoring, besides configurations, fabrication techniques, advantages and disadvantages. The topic is worthy of investigation, but the manuscript suffers from the following shortcomings:
- The research question and has not been put forward clearly.
- Both the level and context of the introduction would benefit researchers who have just started working in this subject. So, it is recommended to separate Introduction from Related Studies and put them in separated section.
- Tables 1, 2 and 3 would be beneficial if get them merged in one comprehensive table just before the Discussion and Conclusion section. Most importantly if could highlight open challenges issues with the existing work.
- All figures should be revised from size and resolution perspectives.
- Figures 1 and 2 do not add any value to the manuscript. Should be removed.
- It is recommended to add subsection about the Forth Industrial Revolution technologies including IoT, UAV, and AI, then state how it effects sustainable agriculture. Here are few references to consider:
https://books.google.com.sa/books?hl=en&lr=&id=LKsXEAAAQBAJ&oi=fnd&pg=PT9&dq=smart+agriculture+iot+4IR&ots=MuogzHtR25&sig=fkzXlt6U0UUcqZDy2ttMdU1rrQA&redir_esc=y#v=onepage&q=smart%20agriculture%20iot%204IR&f=false
https://ieeexplore.ieee.org/abstract/document/9372149
https://www.mdpi.com/2071-1050/13/11/5908
- Discussion and Conclusion section could be revised to include recommendation and future research directions.
- Overall, the manuscript suffers from some issues, so it is recommended to be minor revision.
Author Response
Dear reviewer,
We would like to thank you for your useful comments on our manuscript. Please see the attachment for a point-to-point reply of your comments.

Round 2
Reviewer 2 Report
The authors detail all the changes that I suggested to improve the manuscript. The list of points for to be consider by the authors has been answer. Then, it can be accepted.